# Continual Generalized Category Discovery:
# Learning and Forgetting from a Bayesian Perspective

**Hao Dai** [1 2]  **Jagmohan Chauhan** [1 2]

## Abstract

Continual Generalized Category Discovery (C-GCD) faces a critical challenge: incrementally learning new classes from unlabeled data streams while preserving knowledge of old classes. Existing methods struggle with catastrophic forgetting, especially when unlabeled data mixes known and novel categories. We address this by analyzing C-GCD's forgetting dynamics through a Bayesian lens, revealing that covariance misalignment between old and new classes drives performance degradation. Building on this insight, we propose Variational Bayes C-GCD (VB-CGCD), a novel framework that integrates variational inference with covariance-aware nearest-class-mean classification. VB-CGCD adaptively aligns class distributions while suppressing pseudo-label noise via stochastic variational updates. Experiments show VB-CGCD surpasses prior art by $+15.21\%$ with the overall accuracy in the final session on standard benchmarks. We also introduce a new challenging benchmark with only $10\%$ labeled data and extended online phases—VB-CGCD achieves a $67.86\%$ final accuracy, significantly higher than state-of-the-art ($38.55\%$), demonstrating its robust applicability across diverse scenarios. Code is available at: `https://github.com/daihao42/VB-CGCD`

## 1. Introduction

Continual learning (CL) seeks to address one of the most pressing challenges in machine learning: enabling models to incrementally acquire knowledge from a continuous stream of tasks without succumbing to catastrophic forget-ting—a phenomenon where previously learned information is overwritten (Wang et al., 2024; Rebuffi et al., 2016; Kirkpatrick et al., 2016). This capability is essential in dynamic, real-world applications such as lifelong personal assistants, adaptive robotics, and autonomous systems, where tasks and categories evolve over time. Continual Category Discovery (CCD) (Zhou et al., 2024b) builds on this foundation, tackling the question of how to discover and learn new categories from streaming unlabeled data while retaining the ability to classify previously learned categories. However, real-world data streams often present a more complex challenge: they contain a mix of known and unknown classes, rather than being neatly partitioned into entirely new or entirely old categories (Geng et al., 2021).

C-GCD (Zhang et al., 2022; Wen et al., 2023; Wu et al., 2023) aims to push CL closer to real-world applicability by addressing the challenge of mixed-class data streams. This paradigm raises two critical questions: (1) How can a model differentiate between known and unknown classes in an interwoven data stream without misclassifying known samples? (2) How can a model effectively balance learning new classes while preserving old knowledge within noises? These challenges underscore the delicate trade-off between *stability*, the ability to preserve previously learned knowledge, and *plasticity*, the capacity to adapt to new categories. Existing methods address this tension through contrasting strategies: some prioritize plasticity (Zhang et al., 2022; Rypesc et al., 2024), focusing on learning new classes while applying knowledge distillation or constraints to retain old class representations; while others prioritize stability (Douillard et al., 2022; Hu et al., 2023; Cendra et al., 2024), freezing feature extractors trained on earlier tasks and incrementally expanding classifiers.

To maintain stability, recent approaches have increasingly leveraged pre-trained models (PTMs) (Zhou et al., 2024a), which provide robust feature representations and strong generalization capabilities derived from extensive self-supervised training on diverse datasets. By fine-tuning PTMs, the issue of feature drift can be mitigated to some extent. Many studies have focused on improving new-class learning, such as optimizing inter-class distances (Wu et al., 2023) or extracting category-specific feature representa-

---

[1]Department of Computer Science, UCL Centre for Artificial Intelligence, University College London, London, UK [2]University of Southampton, Southampton, UK. Correspondence to: Hao Dai <daihaovigg@gmail.com>.

*Proceedings of the $42^{nd}$ International Conference on Machine Learning*, Vancouver, Canada. PMLR 267, 2025. Copyright 2025 by the author(s).

tions (Han et al., 2022). While these advancements have significantly improved stability, current PTM-based approaches often mitigate forgetting by introducing regularization to restrict the learning process, which limits their flexibility and adaptability when modeling new and emerging classes.

Motivated by this limitation, we revisited the relationship between learning new classes and forgetting to explore a more dynamic solution. To this end, we propose a Bayesian framework that leverages Gaussian distribution for modeling dynamically evolving categories. We examine the factors influencing classification performance, with a particular focus on the critical role of covariance in balancing the trade-off between stability and plasticity. Our contributions can be summarized as follows:

- We formalize the role of covariance dynamics in C-GCD's forgetting-learning tradeoff, revealing that distribution misalignment (measured via Bhattacharyya distance) irreversibly biases decision boundaries.
- We propose VB-CGCD, a Variational Bayes framework with Mahalanobis distance for C-GCD. By incorporating covariance constraints, our approach enables robust learning of new categories while preserving previously acquired knowledge efficiently.
- We validate VB-CGCD on multiple datasets, showing an average $15.21\%$ improvement in final accuracy over SOTA. We also introduce a label-limited benchmark ($10\%$ categories labeled, 9 online sessions), where VB-CGCD's final accuracy reached $67.86\%$, outperforming the SOTA accuracy of $38.55\%$

## 2. Related Work

**Novel Category Discovery.** New Class Discovery (NCD) aims to identify new, unlabeled classes by leveraging metrics or evaluation criteria learned from known data. For instance, the rank statistics method introduced by AutoNovel (Han et al., 2022) labels new data by sorting features and employing a BCELoss-based learning framework. Similarly, meta-learning approaches like MetaGCD (Wu et al., 2023) have garnered attention for their ability to generalize across tasks by learning class-agnostic similarity functions. These methods rely heavily on effective feature representations and robust distance functions to handle noise and ensure accurate differentiation between classes.

**Generalized Category Discovery.** Generalized Category Discovery (GCD) builds upon NCD by addressing more complex scenarios where unlabeled data contains a mix of new and previously known categories. A central challenge in GCD is distinguishing between known and unknown data within unlabeled datasets. To tackle this, many approaches leverage supervised and unsupervised contrastive learning (Chen et al., 2020) to enhance feature representa-

tions, facilitating the separation of known and novel categories. Additionally, semi-supervised learning (Rizve et al., 2021) has been integrated to group data into meaningful categories, further improving the discovery of new classes.

**Continual Generalized Category Discovery.** GCD typically assumes simultaneous access to labeled data for known classes and unlabeled data for new classes. Continual Generalized Category Discovery (C-GCD) builds on this by integrating the challenges of continual learning and GCD. Specifically, C-GCD requires models to incrementally learn from task streams while discovering new classes and retaining knowledge of previously learned ones. This combines the complexities of NCD and GCD and draws on techniques such as feature representation, contrastive learning, and metric learning.

A unique challenge of C-GCD lies in balancing the learning of new categories with the retention of prior knowledge, especially in exemplar-free settings where raw data from past tasks is unavailable. To address this, many methods rely on auxiliary information, such as prototypes (Snell et al., 2017; Belouadah & Popescu, 2019), attention maps (Saha et al., 2021), or leverage holistic representations of class distributions. CPP (Li et al., 2024) uses class means as prototypes (prompts) and integrates a Transformer as the classifier, while class means are often susceptible to the curse of dimensionality. Among these, Gaussian distribution has emerged as a powerful tool for modeling class distributions. Some approaches (Zhu et al., 2021) use multivariate normal distribution (MVN) to generate replay data to mitigate forgetting, such as Happy (Ma et al., 2024) (which directly estimates distributions) and OCM (Ye & Bors, 2022) (which uses a VAE to approximate Gaussian distributions). On the other hand, PromptCCD (Cendra et al., 2024), utilize Gaussian Mixture Models (GMM) as a prompting mechanism to enhance classification performance. FeCAM (Goswami et al., 2023) directly estimates Gaussian prototypes using statistical methods and performs classification via a nearest-neighbor approach based on these prototypes.

However, these methods often focus exclusively on either reducing forgetting or improving learning, lacking a unified analysis of the interplay between the two. VCL (Nguyen et al., 2018), GVCL (Loo et al., 2021), VAR-GP (Kapoor et al., 2021), and S-FSVI (Rudner et al., 2022) are methods that approach continual learning as a sequence of tasks, utilizing variational inference to regularize parameter updates via the KL divergence. These techniques bridge Bayesian inference with continual learning, providing insights into the trade-off between plasticity and stability. Although these approaches employ variational inference to regularize parameter updates for continual learning, they generally overlook the specific interplay between old and new classes in classification-focused CGCD scenarios.

To address this gap, we propose directly modeling the distribution of each class and analyzing the relationship between forgetting and learning from a Bayesian perspective. This approach provides insights into optimizing overall classification accuracy, paving the way for efficient continual learning capable of adapting to dynamic, real-world scenarios.

## 3. Methodology

### 3.1. Problem Formulation

C-GCD involves two parts of datasets: a labeled dataset $\mathcal{D}_l = \{(\mathbf{x}_i^l, y_i^l)\}$ containing $\mathcal{C}_l$ categories, and an unlabeled dataset $\mathcal{D}_u = \{\mathbf{x}_i^u\}$, encompassing $\mathcal{C}_u$ categories. The unlabeled data is introduced incrementally over $T$ sessions $\{\mathcal{S}_t \mid t = 1, \dots, T\}$, requiring the model to learn continuously in an online fashion. Crucially, the unlabeled data contains both novel categories $\mathcal{C}_n$ and instances from previously seen categories, such that $\mathcal{C}_u = \mathcal{C}_l + \mathcal{C}_n$. This assumption closely mirrors real-world scenarios, where streams of data often mix known and novel categories.

To simplify the analysis, we assume that each session introduces a fixed and identical number of new categories $\mathcal{C}_n^t$. This assumption does not limit the generality of our method, as it can be seamlessly extended to handle varying numbers of new categories. When the number of new categories is unknown, existing approaches (Ronen et al., 2022; Zhao et al., 2023; Cendra et al., 2024) can estimate it effectively. We implemented an extension for unknown new categories counts based on the Silhouette Score; please refer to the Appendix D.1 for experimental details.

The primary goal of C-GCD is to maximize classification accuracy across all sessions while maintaining or improving performance on previously learned categories. Formally, let the classification model be denoted as $\mathcal{F}$ with parameter $\Theta$. Given an input $\mathbf{x}_i^u$, the predicted output is $\tilde{y}_i = \mathcal{F}(\mathbf{x}_i^u; \Theta)$. The classification loss $\mathcal{L}$ measures the discrepancy between the predicted label and the true label. An intuitive solution is to constrain the loss associated with learning new classes so that it does not degrade the loss on previously learned classes, as exemplified by Gradient Episodic Memory (GEM) (Lopez-Paz & Ranzato, 2017). At session $\mathcal{S}_t$, the optimization objective of GEM is to minimize the classification loss over the dataset $\mathcal{D}_u^t$ as follows:

$$\Theta^t = \underset{\Theta \in H}{\arg\min} \sum_{\mathbf{x}_i^u \in \mathcal{D}_u^t} \mathcal{L}(\mathcal{F}(\mathbf{x}_i^u; \Theta^t))$$

$$s.t. \sum_{\mathbf{x}_i \in \mathcal{D}^{0:t-1}} \mathcal{L}(\mathcal{F}(\mathbf{x}_i; \Theta^t)) \leq \sum_{\mathbf{x}_i \in \mathcal{D}^{0:t-1}} \mathcal{L}(\mathcal{F}(\mathbf{x}_i; \Theta^{t-1}))$$

$$t = \{1, \dots, T\}$$

$$(1)$$

### 3.2. Forgetting and Learning in C-GCD

Forgetting in C-GCD arises from various factors, the primary challenges are associated with label bias, label error, and trade-off between learning and forgetting.

#### 3.2.1. CHALLENGE 1: LABEL BIAS

Conventionally, classification tasks rely on optimizing the maximum likelihood function to model $\mathbb{P}(y_i|\mathbf{x}_i)$. However, in the continual learning setting, labels are introduced sequentially across different online sessions, leading to the issue of label bias. This bias arises because the model tends to overfit to the newly observed categories after training on a new dataset, often misclassifying samples into these newly learned classes, that is, the knowledge related to old classes is forgotten. This forgetting manifests in two key aspects: (1) Feature Space Shift: Continuous updates to the backbone network cause shifts in the feature distribution, resulting in the failure of the old classification memory. (2) Class Distribution Imbalance: Imbalanced category distributions across sessions lead the classifier to assign disproportionately higher likelihood probabilities to the new classes.

*Self-Supervised Learning based Feature Extractor.* To avoid feature shift, many studies (Cha et al., 2021; Han et al., 2022; Zhang et al., 2022; Ma et al., 2024) have introduced self-supervised learning approaches. These methods learn label-independent features from the data through techniques such as data augmentation, thereby mitigating bias. For example, by optimizing a contrastive loss:

$$\mathcal{L}_{con} = -\frac{1}{\|\mathcal{D}\|} \log \frac{\exp(\langle h(\mathbf{x}_i) \rangle \cdot \langle h(\mathbf{x}_i') \rangle / \tau)}{\sum_{i \neq j}^{\|\mathcal{D}\|} \exp(\langle h(\mathbf{x}_i) \rangle \cdot \langle h(\mathbf{x}_j) \rangle / \tau)} \quad (2)$$

here, $h(\cdot)$ denotes a label-agnostic feature extraction function ($\mathbb{R}^N \to \mathbb{R}^M$). With the rapid development of Vision Transformers (ViTs), numerous self-supervised pre-trained models, such as DINO (Caron et al., 2021), have been integrated into continual learning frameworks, effectively reducing feature-level label bias.

Previous methods (Zhang et al., 2022; Ma et al., 2024), while leveraging pre-trained models, continued to use self-supervised loss during continual learning and incorporated cross-entropy loss with pseudo-labels as supervision signals. This approach introduces the following challenges: (1) Feature drift remains unresolved; (2) Errors introduced by pseudo-labels exacerbate the drift; (3) The training process becomes more burdensome, with redundant forward and backward passes consuming significant training time (Shi et al., 2024).

**Offline Fine-Tuning.** To mitigate feature drift, we perform fine-tuning exclusively during the offline stages and

keep the backbone network parameters frozen throughout all sessions, including offline and online classification learning. This decoupling of fine-tuning and continual learning eliminates the burden of extensive backbone training and preserves the stability of the feature distribution.

It is anticipated that incorporating correct labels to fine-tune the backbone network can enhance feature discriminability, leading to improved classification performance. Fortunately, the labeled data available from the offline phase can be leveraged for this purpose. Combined with the contrastive and cross-entropy loss, the fine-tuning loss is expressed as:

$$\mathcal{L}_{ft} = (1 - \lambda)\mathcal{L}_{ce} + \lambda\mathcal{L}_{con} \tag{3}$$

*Generative Classifier.* To mitigate label bias caused by class imbalance, we adopt a generative classifier that directly models class-conditional feature distributions, which involves modeling $\mathbb{P}(\mathbf{x}_i|y_i)$ instead of learning $\mathbb{P}(y_i|\mathbf{x}_i)$. A representative example of this paradigm is prototypical learning (Snell et al., 2017), which learns a prototype for each class by estimating the class mean $c_k \in \mathbb{R}^M$. Building on a predefined distance function $d := \mathbb{R}^M \times \mathbb{R}^M \to [0, +\infty)$, this approach employs the nearest class mean (NCM) method to estimate the likelihood:

$$\mathbb{P}(y_i = k|\mathbf{x}_i) = \frac{\exp(-d(h(\mathbf{x}_i), c_k))}{\sum_{k'} \exp(-d(h(\mathbf{x}_i), c_{k'}))} \tag{4}$$

However, this method suffers the curse of dimensionality. As data distribution becomes increasingly sparse in high-dimensional spaces, Euclidean distances tend to concentrate, rendering classification failure.

**MVN-based NCM.** To alleviate the issue of classification failure in high dimensions, kernel functions can be employed to collapse the high-dimensional hypersphere. Therefore, we extend NCM by modeling each class as a multivariate normal distribution (MVN), denoted as $\mathcal{N}(\mu_k, \Sigma_k)$, estimated via:

$$\begin{aligned} \mu_k &= \frac{1}{\|\mathcal{D}_k\|} \sum_{\mathbf{x}_i \in \mathcal{D}_k} h(\mathbf{x}_i) \\ \Sigma_k &= \frac{1}{\|\mathcal{D}_k\|} \sum_{\mathbf{x}_i \in \mathcal{D}_k} (h(\mathbf{x}_i) - \mu_k)(h(\mathbf{x}_i) - \mu_k)^T \end{aligned} \tag{5}$$

This captures inter-feature correlations critical in high-dimensional spaces. The distance metric then generalizes to the Mahalanobis similarity:

$$d(h(\mathbf{x}_i), c_k) = \exp(-\frac{1}{2}(h(\mathbf{x}_i) - \mu_k)^T \Sigma_k^{-1} (h(\mathbf{x}_i) - \mu_k)) \tag{6}$$

which adapts to the data's covariance structure, addressing the curse of dimensionality. Under Gaussian class-conditional assumptions, MVN-NCM classifier is Bayes-optimal, with the Mahalanobis term in $d(\cdot)$ acting as the

quadratic discriminant (Goswami et al., 2023). While mitigating label bias, this method provides inherent robustness—outliers yield low likelihoods, while $\Sigma_k^{-1}$ downweights noisy features.

### 3.2.2. CHALLENGE 2: LABEL ERRORS

In the C-GCD scenario, where data is unlabeled, novelty learning relies on pseudo-labels generated by algorithms such as clustering. However, this process inevitably introduces substantial label errors (misclassifications). The error originates from two primary sources: (1) interference from data belonging to known classes, which disrupts the generation of accurate pseudo-labels for new classes, and (2) inherent errors in the clustering algorithm.

*Distinguishing between New and Old Categories.* Effectively separating unlabeled samples into new and known classes is crucial to minimizing interference. While likelihood-based methods often struggle with this distinction, distance-based approaches (Zhang et al., 2022) classify samples by applying a distance threshold. However, these methods rely heavily on arbitrary cutoffs, which can significantly hinder generalizability.

**Re-Labeling.** To address this, we propose a self-correcting re-labeling mechanism that operates in two stages: (1) Pseudo-Label Initialization: Unlabeled data is clustered to assign provisional labels, forming an initial estimate of the new-class distribution $\mathcal{N}(\mu'_k, \Sigma'_k)$. (2) Classifier Merging and Refinement: The latent variables of old classes and the pseudo-labeled distribution are merged into a unified MVN-NCM classifier, which then refines label assignments for the unlabeled data.

The pseudo-labels assigned by the clustering algorithm inevitably contain a substantial amount of noise, causing the fitted distribution $\mathcal{N}(\mu'_k, \Sigma'_k)$ to deviate significantly from the old classes' distributions. Consequently, data originally belonging to old classes remains naturally closer to the old distribution, making it more likely to be reassigned to its correct label (old). At the same time, since the fitted distribution is closer to the true new-class distribution $\mathcal{N}(\mu^*_k, \Sigma^*_k)$, new-class samples tend to align more closely with the fitted distribution than with old classes' distributions, preserving their pseudo labels.

*Noise-Robust Distribution Fitting.* While re-labeling helps reduce known class noise, clustering errors inevitably propagate into the estimated distributions when using a statistical approach, as shown in Equation (5). As a result, a noise-robust, learnable estimation method is necessary to effectively fit $\mathbb{P}(\mathbf{x})$. However, in most cases, the marginal distribution $\mathbb{P}(\mathbf{x})$ is a non-analytic integral and does not have a closed-form solution.

**Variational Bayes MVN.** To approximate the marginal

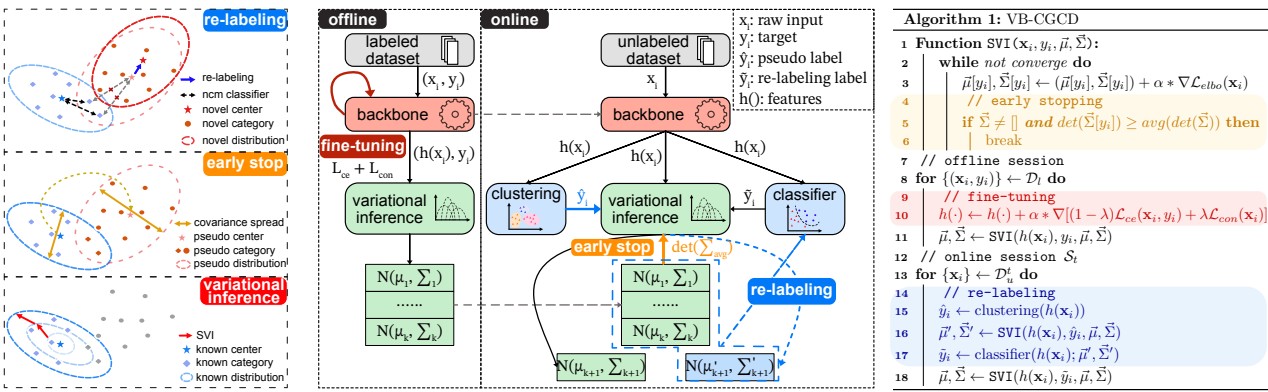

*Figure 1.* The overall architecture of VB-CGCD. Offline Stage: Labeled data is used to fine-tune the backbone network. After fine-tuning, features are extracted, and variational inference is performed to obtain and store the class distribution. Online Stage: Unlabeled data is first processed by the feature network to extract features, which are then clustered to assign pseudo-labels. These pseudo-labels are combined with a first round of variational inference to obtain an approximate distribution, then combined with the stored class distribution from the previous session for re-labeling.

distribution of $\mathbf{x}$, we propose the Variational Bayes-based Multivariate Normal distribution (VB-MVN) to estimate the posterior distribution. We treat the means and covariances in posterior distribution $\mathcal{N}(\mu_k, \Sigma_k)$ as latent variable $z$, and approximate the posterior $p_\Phi(z|\mathbf{x})$ with a variational distribution $q_\Theta(z|\mathbf{x})$. This yields the Evidence Lower Bound (ELBO) (Kingma & Welling, 2014):

$$\mathcal{L}_{ELBO} = \mathbb{E}_{q_\Theta(z|\mathbf{x})}[\ln p_\Phi(\mathbf{x}|z)] - \mathbb{D}_{KL}(q_\Theta(z|\mathbf{x})||p_\Phi(z)) \tag{7}$$

where $p_\Phi(z)$ initialized to $\mathcal{N}(\vec{0}, \mathbb{I})$ acts as a conjugate prior, ensuring closed-form updates.

We optimize ELBO via stochastic variational inference (SVI) (Hoffman et al., 2012), which provides two critical advantages: (1) Outlier Suppression: Mislabeled samples exhibit low likelihood, contributing minimally to the reconstruction term. Simultaneously, their deviation from the distribution of $z$ incurs a strong Kullback-Leibler (KL) penalty, effectively downweighting their influence. (2) Scalability: SVI's stochastic gradients enable efficient optimization on large-scale data, avoiding the cost of full-batch inference.

### 3.2.3. CHALLENGE 3: TRADE-OFF BETWEEN LEARNING AND FORGETTING.

Suppose the old model captures the distributions $\{\mathcal{N}(\mu_i, \Sigma_i) \mid i = 1, \dots, k-1\}$ of $k-1$ known classes. When learning a new category $c_k$, the prior is set to $\mathcal{N}(\vec{0}, \mathbb{I})$, a zero-mean isotropic Gaussian, to avoid premature bias toward existing classes while ensuring numerical stability during early training phases. By design, the test dataset initially exhibits stronger similarity (Equation 6) to the distributions of known classes than to the new class, preserving the old model's accuracy (Figure 2(a), P1).

However, as learning progresses and the accuracy of the new class continues to improve, the accuracy of the old classes begins to experience a slight decline (P2). Beyond a certain point, the continued improvement of new class accuracy comes at a cost: the overall accuracy progressively decreases, marking the onset of catastrophic forgetting (P3). Restricting the decline in old class accuracy to accommodate the learning of new classes may seem like an intuitive solution. However, as depicted by the black dashed line in Figure 2(a), such approaches often lead to suboptimal performance.

*Covariance Impacts Forgetting.* As variational inference iterates on, the posterior variance $\Sigma_k$ monotonically increases (Figure 2(b)), diverging from $\mathbb{I}$ and eventually exceeding the variance of older classes. This divergence arises because ELBO naturally penalizes over-regularization, allowing $\Sigma_k$ to expand until it sufficiently explains the new data's covariance. From the perspective of the NCM classifier, when assuming the covariance matrix is diagonal, the distance function can be expressed as:

$$d(h(\mathbf{x}_i), c_k) \propto \sum_j^M \frac{(\mathbf{x}_i^j - \mu_k^j)^2}{(\sigma_k^j)^2} \tag{8}$$

The distributional difference between new and old classes can be quantified using the Bhattacharyya distance:

$$d_B(p_1, p_2) = \frac{1}{8}(\mu_2 - \mu_1)^T (\frac{\Sigma_1 + \Sigma_2}{2})^{-1}(\mu_2 - \mu_1) + \frac{1}{2}\ln\frac{\det(\frac{\Sigma_1 + \Sigma_2}{2})}{\sqrt{\det(\Sigma_1)\det(\Sigma_2)}} \tag{9}$$

The three stages depicted in Figure 2(a) can be explained via covariance as follows:

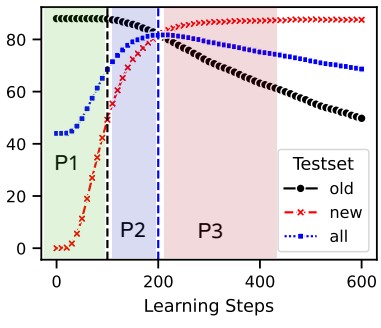

(a) Accuracy

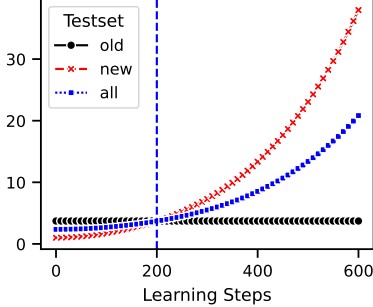

(b) Determinants of covs

*Figure 2.* (a) The evolution of accuracy for old classes, new classes, and overall during the learning process of online session. (b) The variation in the average determinants of the covariance.

- P1 (Stability): The new-class accuracy improves while old-class accuracy remains intact, as the initial prior minimally perturbs existing distributions.
- P2 (Critical Transition): As $\Sigma_k$ expands, the new class's Mahalanobis distance (Equation (8)) becomes increasingly weighted by features' variance, which induces overlap between old and new classes, causing a slight decline in old-class accuracy.
- P3 (Forgetting): When $\Sigma_k$ exceeds the variance of old classes, $d_B$ grows asymmetrically, skewing the decision boundary toward the new class. Old-class accuracy degrades irreversibly, despite new-class accuracy stabilizing.

The root cause of forgetting lies in the covariance mismatch. According to Equation (8), the distance metric $d(\cdot)$ applies soft feature selection by weighting dimensions inversely by $\sigma_k^j$. As covariance grows, the method increasingly ignores high-variance (noisy) features of the new class, effectively narrowing its focus to stable, discriminative features. While this improves new-class accuracy, it also: (1) distorts the Mahalanobis space for old classes, whose fixed $\Sigma_{1:k-1}$ no longer align with the new feature weighting scheme; (2) diases samples toward the new class, when $\Sigma_k \gg \Sigma_{1:k-1}$, the second term in Equation (9) dominates, inflating $d_B$ and fragmenting the decision boundary.

**Early Stopping.** To mitigate catastrophic forgetting while maximizing new-class plasticity, we formalize a critical in-

sight: optimal classification performance requires balancing the covariance scales of old and new classes. According to Equation (9), we propose a regularization term to align covariance determinants by penalizing deviations from the old classes' average covariance:

$$\mathcal{L}_{det} = (\log \det(\Sigma_k) - \log \frac{1}{k-1} \sum_{i=1}^{k-1} \det(\Sigma_i))^2 \quad (10)$$

However, naively optimizing alongside ELBO fails in practice due to: (1) ELBO's inherent bias: The variational objective prioritizes data likelihood over regularization, leaving it under-optimized. (2) Data asymmetry: In the online phase, the number of samples and categories differs significantly from the offline phase, resulting in complex hyperparameter tuning. Instead, we propose an early stopping rule that halts training when new-class covariances align with old ones. Specifically, we monitor the ratio:

$$\mathcal{R} = \log \frac{(k-1) \det(\Sigma_k)}{\sum_{i=1}^{k-1} \det(\Sigma_i)} \quad (11)$$

terminating training when $\mathcal{R} < \epsilon$ (empirically, $\epsilon = 0.01$), effectively minimizing $\mathcal{L}_{det}$. When $\Sigma_k$ approach to $\Sigma_{1:k-1}$, we have: (1) the Bhattacharyya distance is minimized, yielding the most compact and separable decision boundary; (2) feature weighting balances stability (old classes) and plasticity (new class), avoiding bias toward either. As illustrated in Figure 2(b), the intersection of the covariances marks the point where the overall accuracy reaches its peak.

### 3.3. Variational Bayes C-GCD

As outlined in Table 1, we propose a method called **VB-CGCD** (**V**ariational **B**ayes **C-GCD**), addressing label bias, pseudo-label errors, and the learning-forgetting tradeoff through these core components: (1) Self-supervised learning based offline fine-tuning. (2) Self-corrective re-labeling. (3) Variational Bayesian distribution estimation and nearest class mean classifier. (4) Covariance-driven early stopping.

*Table 1.* Summary of the components of VB-CGCD.

|  | Label Bias | Label Errors | Trade-Off |
|---|---|---|---|
| Offline Fine-Tuning | ✓ | - | - |
| Re-Labeling | ✗ | ✓ | ✗ |
| VB-MVN + NCM | ✓ | ✓ | ✗ |
| Early-Stopping | ✗ | ✗ | ✓ |

Figure 1 illustrates VB-CGCD's two-phase design: *Offline Phase*: (1) Fine-tune the backbone network on initial labeled data; (2) Fit labeled class-conditional distributions via SVI. *Online Phase*: (1) Assign pseudo-labels to unlabeled data through clustering, then refine them via re-labeling. (2) Fit distributions via SVI, constrained by covariance-aware early stopping.

*Table 2.* Performance comparison on Cifar100 (C100), ImageNet100 (IN100), TinyImageNet (Tiny), and CUB200 (CUB).

| Datasets | Methods | $\mathcal{S}_0$ | $\mathcal{S}_1$ | | | $\mathcal{S}_2$ | | | $\mathcal{S}_3$ | | | $\mathcal{S}_4$ | | | $\mathcal{S}_5$ | | |
|---|---|---|---|---|---|---|---|---|---|---|---|---|---|---|---|---|---|
| | | All | All | Old | New | All | Old | New | All | Old | New | All | Old | New | **All** | Old | New |
| C100 | GM | 90.36 | 76.58 | 79.80 | 60.50 | 71.10 | 74.52 | 50.60 | 63.51 | 68.16 | 31.00 | 59.74 | 62.51 | 37.60 | 54.11 | 54.74 | 48.40 |
| | MetaGCD | 90.82 | 76.12 | 83.60 | 38.70 | 69.40 | 72.82 | 48.90 | 61.95 | 65.76 | 35.30 | 58.22 | 61.21 | 34.30 | 55.78 | 58.47 | 31.60 |
| | PromptCCD | 85.20 | 63.71 | 65.03 | 57.13 | 61.80 | 62.86 | 55.46 | 60.14 | 61.81 | 53.54 | 52.65 | 52.89 | 50.72 | 55.53 | 55.64 | 54.54 |
| | Happy | 90.10 | 83.62 | 85.08 | 76.30 | 75.83 | 80.85 | 45.70 | 70.03 | 73.50 | 45.70 | 64.70 | 65.06 | 61.80 | 60.77 | 62.02 | 49.50 |
| | VB-CGCD | **91.68** | **88.40** | **89.46** | **83.10** | **86.0** | **87.06** | **79.60** | **83.61** | **84.75** | **75.60** | **82.72** | **82.85** | **81.70** | **81.23** | **81.84** | **75.70** |
| Tiny | GM | 85.86 | 76.42 | 82.40 | 46.50 | 68.87 | 73.82 | 39.20 | 58.68 | 63.43 | 25.40 | 52.86 | 57.21 | 18.10 | 46.90 | 50.62 | 13.40 |
| | MetaGCD | 84.20 | 60.88 | 64.90 | 40.80 | 57.20 | 61.03 | 34.20 | 54.36 | 57.19 | 34.60 | 50.83 | 53.59 | 28.80 | 48.14 | 50.16 | 30.00 |
| | PromptCCD | 79.23 | 64.10 | 64.72 | 61.02 | 61.43 | 62.07 | 57.58 | 56.39 | 57.08 | 51.53 | 55.38 | 55.52 | 54.27 | 50.74 | 50.82 | 50.0 |
| | Happy | 85.76 | 77.80 | 82.68 | 53.40 | 72.26 | 76.02 | 49.70 | 65.29 | 70.06 | 31.90 | 57.79 | 61.40 | 28.90 | 54.93 | 56.08 | 44.60 |
| | VB-CGCD | **88.32** | **85.10** | **86.44** | **78.40** | **82.40** | **84.15** | **71.90** | **79.35** | **81.38** | **65.10** | **76.72** | **78.43** | **63.0** | **74.52** | **75.88** | **62.20** |
| IN100 | GM | 96.20 | 89.53 | 95.04 | 62.00 | 82.34 | 86.93 | 54.80 | 77.97 | 79.17 | 69.60 | 72.80 | 74.65 | 58.00 | 71.08 | 71.76 | 65.00 |
| | MetaGCD | 95.96 | 75.27 | 78.20 | 60.60 | 73.79 | 75.93 | 54.90 | 69.35 | 72.20 | 49.40 | 67.22 | 70.10 | 44.20 | 66.68 | 69.31 | 43.00 |
| | PromptCCD | 91.18 | 82.11 | 82.36 | 80.84 | 80.03 | 80.36 | 78.02 | 77.78 | 78.03 | 76.05 | 67.37 | 67.54 | 65.99 | 63.97 | 64.08 | 62.95 |
| | Happy | 96.20 | 91.20 | 95.36 | 70.40 | 87.83 | 90.83 | 69.80 | 85.22 | 86.40 | 77.00 | 81.93 | 83.00 | 73.40 | 78.58 | 79.11 | 73.80 |
| | VB-CGCD | 94.32 | **93.13** | 93.16 | **93.0** | **90.45** | **91.43** | **84.60** | **90.10** | **90.0** | **90.8** | **88.97** | **89.30** | **86.40** | **87.32** | **88.08** | **80.40** |
| CUB | GM | 90.26 | 76.17 | 80.23 | 56.51 | 67.91 | 73.38 | 34.58 | 61.12 | 66.53 | 23.00 | 55.90 | 57.49 | 43.38 | 51.96 | 54.40 | 30.10 |
| | MetaGCD | 89.20 | 67.08 | 70.21 | 51.92 | 60.77 | 62.39 | 50.86 | 57.53 | 59.33 | 37.78 | 51.90 | 52.22 | 49.40 | 49.60 | 49.96 | 46.38 |
| | PromptCCD | 82.10 | 63.21 | 63.91 | 59.71 | 51.0 | 52.17 | 43.98 | 51.80 | 52.57 | 41.43 | 46.21 | 46.86 | 41.61 | 50.95 | 51.16 | 49.06 |
| | Happy | 86.0 | 77.83 | 82.44 | 55.37 | 68.19 | 76.24 | 21.51 | 63.44 | 67.47 | 34.63 | 58.39 | 61.90 | 31.14 | 54.37 | 56.70 | 33.67 |
| | VB-CGCD | 85.72 | **78.46** | **82.92** | **55.45** | **74.60** | **76.72** | **62.01** | **71.84** | **73.38** | **60.83** | **68.60** | **70.91** | **49.64** | **66.44** | **67.76** | **54.38** |

# 4. Experiments

We evaluated VB-CGCD on a variety of datasets. Additionally, we conducted comparisons with several baseline approaches to assess its performance.

## 4.1. Experimental Setup

**Datasets.** In line with mainstream C-GCD works, we utilized four datasets—CIFAR100 (Krizhevsky, 2009), TinyImageNet (Le & Yang, 2015), ImageNet100 (Russakovsky et al., 2015), and CUB200 (Wah et al., 2011)—each representing distinct characteristics. To ensure fair comparisons, we followed the experimental setup described in (Ma et al., 2024):

*Training data*: The labeled data consists of 50% of the classes, with 80% of the training samples per class. The unlabeled data comprises the remaining 50% of the classes, evenly distributed across all online sessions, with 80% of the training samples per class. Additionally, 20% of the data from all previously encountered classes is carried forward into each session.

*Testing data*: In each session, the test set includes samples from all classes encountered up to that session.

**Baselines.** We compared VB-CGCD with several state-of-the-art approaches, including *MetaGCD* (Wu et al., 2023), *GM* (Zhang et al., 2022), *PromptCCD* (Cendra et al., 2024), *Happy* (Ma et al., 2024).

**Evaluation Metrics.** We employed the following metrics to evaluate the performance of VB-CGCD: (1) the accuracy of

newly introduced classes in the current session ('New'), the accuracy of all previously encountered classes ('Old'), and the accuracy of all classes currently ('All'). (2) Forgetting rate $\mathcal{M}_f$, the difference in the accuracy of labeled classes between the model in the offline phase and the model after completing all online sessions. (3) Novelty learning rate $\mathcal{M}_d$, the average accuracy of new classes in each session.

More details of the data configuration, metrics, implementation, and hyperparameters, can be found in the Appendix.

## 4.2. Results

**Accuracy.** The overall results are shown in Table 2. When the number of online sessions is set to 5, VB-CGCD consistently outperforms the baselines in overall accuracy, as well as old-class and new-class accuracy. Throughout the continual learning process, VB-CGCD demonstrates strong resistance to forgetting while significantly enhancing performance on novelty learning. As a result, by the end of all continual learning sessions, the final overall accuracies ('All' of $\mathcal{S}_5$) achieved by VB-CGCD exceed those of the baselines across four datasets, with an average improvement of 15.21%.

**Forgetting and Learning.** To further investigate the factors influencing performance, we analyzed the forgetting rate $\mathcal{M}_f$ and novelty learning rate $\mathcal{M}_d$. For comparison, we constructed a performance bound by dividing the dataset into two parts and performing supervised learning on each part separately. This setup eliminates the effects of label error (since all samples are fully labeled) and label bias (which is restricted to only two stages). From this analysis,

Table 3. Forgetting rate and novelty learning rate (↑ indicates that the higher the metric, the better, and vice versa).

| Methods | C100 | | Tiny | | IN100 | | CUB | |
|---|---|---|---|---|---|---|---|---|
| | $\mathcal{M}_f \downarrow$ | $\mathcal{M}_d \uparrow$ | $\mathcal{M}_f \downarrow$ | $\mathcal{M}_d \uparrow$ | $\mathcal{M}_f \downarrow$ | $\mathcal{M}_d \uparrow$ | $\mathcal{M}_f \downarrow$ | $\mathcal{M}_d \uparrow$ |
| PromptCCD | 21.66 | 54.28 | 23.46 | 54.88 | 18.76 | 72.77 | 19.10 | 47.15 |
| Happy | 12.08 | 55.80 | 9.16 | 41.70 | 10.13 | 72.88 | **3.22** | 35.26 |
| VB-CGCD | **6.94** | **79.14** | **5.72** | **68.12** | **5.68** | **87.04** | 6.72 | **56.46** |
| UpperBound | 5.14 | 83.76 | 3.48 | 77.04 | 3.16 | 89.04 | 5.13 | 79.65 |

we infer the following: (1) Surpassing the upper bound of $\mathcal{M}_d$ will likely result in excessive forgetting. (2) Falling below the lower bound of $\mathcal{M}_f$ may lead to inadequate novelty learning.

As shown in Table 3, VB-CGCD achieves performance closer to the bounds for both the forgetting rate and novelty learning rate. On IN100, VB-CGCD nearly reaches the bounds, likely due to the large dataset size (1000 samples per class), which facilitates learning a more accurate distribution. In contrast, on CUB, where each class contains only 30 samples, the limited data exacerbates the effects of label noise, resulting in lower novelty learning performance. Additionally, it is noteworthy that *Happy* falls below the lower bound for forgetting, highlighting its insufficient novelty learning capabilities.

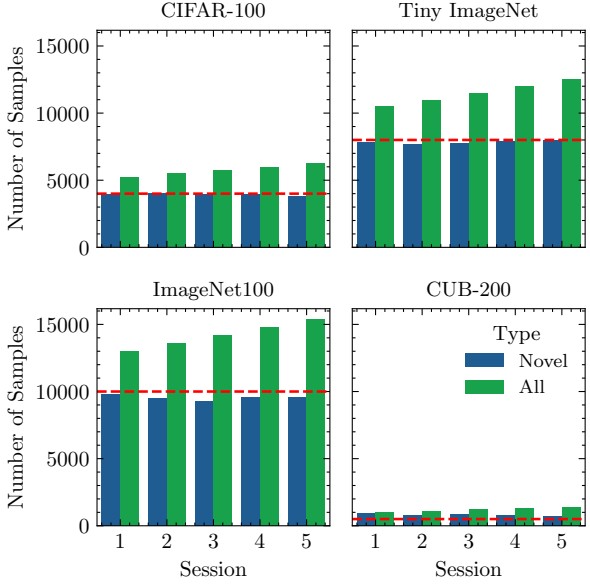

Figure 3. Comparison of the identified number of samples belonging to new classes (denoted as 'Novel') with the total number of samples (denoted as 'All'). The red horizontal dashed line represents the ground truth of 'Novel'.

**Novelty Detection.** Additionally, we tracked the number of samples identified as belonging to new classes in each session, as shown in Figure 3. The re-labeling process effectively distinguished new classes from old ones with high accuracy. For datasets with sufficient samples per class, such as C100, Tiny, and IN100, the accuracy reached approximately 95%. However, on the CUB dataset, which contains a limited number of samples, the mixing of old and new classes posed greater challenges, resulting in a lower accuracy of around 65%. This limitation provides a partial explanation for the relatively lower overall accuracy observed on CUB.

**Ablation Study.** To assess the contribution of each module, we conducted ablation experiments focusing on four key components: SVI, fine-tuning, early stopping, and re-labeling. The experimental configurations, outlined in Table 4, include four distinct setups. Notably, in the absence of SVI, the mean, and covariance are estimated directly using Equation (5) through point estimation—a non-learning-based approach. Consequently, early stopping cannot be applied in this configuration.

Table 4. Setup of the ablation study.

| | SVI | Fine-Tuning | Early-Stop | Re-Labeling |
|---|---|---|---|---|
| w/o SVI† | ✗ | ✓ | ✗ | ✓ |
| w/o fine-tuning | ✓ | ✗ | ✓ | ✓ |
| w/o early-stopping | ✓ | ✓ | ✗ | ✓ |
| w/o re-labeling | ✓ | ✓ | ✓ | ✗ |
| w/ all | ✓ | ✓ | ✓ | ✓ |

The ablation results, presented in Table 5, reveal that SVI has the most significant impact on accuracy. Even during the supervised $\mathcal{S}_0$ session, performance without SVI is substantially worse than with it. This disparity arises because the number of samples per class is smaller than the number of features, resulting in a non-full-rank covariance matrix. This issue prevents the computation of the distance function, making classification ineffective. To address this, dimensionality reduction was applied to the datasets during the experiments, ensuring the results remained meaningful.

In summary, the removal of any module results in performance degradation, with each module contributing uniquely to the overall system. SVI and fine-tuning enhance the correlation between features, aiding both the mitigation of forgetting and the learning of new classes, while early stopping and re-labeling focus on achieving a balance between new-class learning and forgetting. Early stopping prevents the covariance values from exceeding those of known classes, thereby avoiding overfitting to new classes, which would otherwise reduce old-class accuracy and ultimately lower overall accuracy. Without re-labeling, however, significant amounts of noisy data are introduced, reducing the average covariance during training and delaying the early stopping mechanism. Consequently, in experiments without these two methods, the accuracy for new classes is higher, but overall accuracy is diminished.

Among the modules, the contribution of re-labeling to overall ('All') accuracy is relatively minor, yielding approxi-

*Table 5.* Ablation study results across multiple datasets.

| Datasets | Methods | All($\mathcal{S}_0$)↑ | All($\mathcal{S}_5$)↑ | $\mathcal{M}_f$ ↓ | $\mathcal{M}_d$ ↑ |
|---|---|---|---|---|---|
| C100 | w/o SVI† | 88.92 | 68.90 | 17.10 | 72.24 |
| | w/o fine-tuning | 88.54 | 73.48 | 8.18 | 68.0 |
| | w/o early-stopping | 91.68 | 77.71 | 15.8 | **81.62** |
| | w/o re-labeling | 91.68 | 80.20 | 9.58 | 80.74 |
| | w/ all | **91.68** | **81.23** | **6.94** | 79.14 |
| Tiny | w/o SVI† | 84.78 | 61.14 | 14.22 | 56.52 |
| | w/o fine-tuning | 83.94 | 68.69 | 5.74 | 60.62 |
| | w/o early-stopping | 88.32 | 71.23 | 14.24 | **70.60** |
| | w/o re-labeling | 88.32 | 73.67 | 8.48 | 70.38 |
| | w/ all | **88.32** | **74.52** | **5.72** | 68.12 |
| IN100 | w/o SVI† | 92.56 | 79.90 | 8.45 | 80.32 |
| | w/o fine-tuning | 93.04 | 84.0 | 7.36 | 83.88 |
| | w/o early-stopping | 94.32 | 84.96 | 11.92 | **89.04** |
| | w/o re-labeling | 94.32 | 86.54 | 7.16 | 88.28 |
| | w/ all | **94.32** | **87.32** | **5.68** | 87.04 |
| CUB | w/o SVI† | 55.34 | 27.06 | 40.08 | 49.30 |
| | w/o fine-tuning | 80.16 | 51.10 | 15.12 | 40.99 |
| | w/o early-stopping | 85.72 | 56.43 | 29.22 | **60.77** |
| | w/o re-labeling | 85.72 | 63.32 | 13.44 | 59.65 |
| | w/ all | **85.72** | **66.44** | **6.72** | 56.46 |

*Table 6.* Performance of each session with 10-sessions setting.

| Datasets | Methods | $\mathcal{M}_f$ ↓ | $\mathcal{M}_d$ ↑ | All($\mathcal{S}_0$) | All($\mathcal{S}_9$) | b50-All($\mathcal{S}_5$) |
|---|---|---|---|---|---|---|
| C100 | Happy | 36.90 | 48.40 | 98.10 | 36.16 | 60.77 |
| | VB−CGCD | **18.30** | **78.38** | **98.90** | **74.72** | **81.32** |
| TINY | Happy | 66.46 | 33.39 | 88.80 | 22.11 | 54.93 |
| | VB−CGCD | **7.40** | **67.68** | **92.70** | **66.30** | **74.52** |
| IN100 | Happy | **10.0** | 62.15 | 91.40 | 59.12 | 75.58 |
| | VB−CGCD | 14.8 | **85.76** | **98.20** | **82.88** | **87.32** |
| CUB | Happy | **21.07** | 42.87 | 94.17 | 36.80 | 54.37 |
| | VB−CGCD | 28.54 | **49.54** | 94.02 | **47.46** | **66.44** |

mately a $1\%$ improvement. This is because the SVI method effectively mitigates the influence of outliers, ensuring that moderate levels of label noise do not significantly impair the learning process. However, in scenarios with smaller datasets and higher noise ratios—such as CUB—the advantages of re-labeling become more evident, delivering an improvement of around $3\%$.

**Fewer Labeled Categories Scenario.** Previous studies often assume that the majority of old classes are labeled. However, given the ability of VB-CGCD to mitigate label noise and bias, it can be extended to handle more extreme scenarios with a larger proportion of unlabeled classes. To evaluate this capability, we designed an extreme case where $90\%$ of the classes are unlabeled, and distributed across 9 sessions, with only $10\%$ of the classes having partially labeled data ($80\%$ of their samples).

The results, shown in Table 6, highlight the advantages of our method over the current state-of-the-art, *Happy*, in both novelty learning and final accuracy. Notably, on IN100, while *Happy* demonstrates $4.8\%$ less forgetting compared to VB-CGCD, this is offset by the fact that VB-CGCD

achieves $6.8\%$ higher accuracy than *Happy* in the supervised session ($\mathcal{S}_0$), resulting in a net improvement of $2\%$ in $\mathcal{M}_f$. For CUB, the trend remains consistent with earlier results: although *Happy* strictly prevents forgetting, this comes at the expense of new-class accuracy, ultimately leading to lower final accuracy. We also compared VB-CGCD to a baseline with $50\%$ labeled classes ('b50'). Despite reduced labeled categories, VB-CGCD showed only an average $9.56\%$ accuracy drop, far lower than the $22.87\%$ drop in *Happy*, demonstrating its strong generalizability.

## 5. Conclusion and Discussion

This paper presents VB-CGCD, a Bayesian framework for C-GCD that models feature distributions via variational inference and mitigates forgetting through covariance alignment. The study investigates the underlying causes of forgetting within this framework, identifies the critical role of covariance, and proposes an early stopping rule that halts training at optimal stability-plasticity balance. Experimental across diverse benchmarks validate that VB-CGCD significantly outperforms existing baselines, demonstrating robust performance and excellent scalability.

While VB-CGCD demonstrates significant potential, several challenges remain, presenting avenues for future research. (1) Its reliance on initial clustering quality can impact performance, particularly in high-dimensional spaces. Integrating deep clustering techniques or adaptive mixture models could enhance robustness and reduce sensitivity to initialization. (2) Small-sample covariance estimation, as seen in datasets like CUB, remains a limitation. Investigating hierarchical Bayesian priors or meta-learning strategies to improve covariance estimation under limited data conditions could further enhance generalization.

## Acknowledgements

This work was supported by the Engineering and Physical Sciences Research Council (EPSRC) grant, MultiTasking and Continual Learning for Audio Sensing Tasks on Resource-Constrained Platforms [EP/X01200X/1].

## Impact Statement

By leveraging variational inference and covariance constraints, this method addresses catastrophic forgetting and label noise in online learning scenarios. Its robustness, noise resistance, and broad applicability make it a promising solution for more general class-incremental learning tasks.

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

## A. Pseudocode

---

**Algorithm 1:** VB-CGCD

---

**Input**  : steaming datasets $\mathcal{D} = \{\mathcal{D}_l\} \cup \{\mathcal{D}_u^t \mid t = 1, \ldots, T\}$, ViTB-16 $h(\cdot)$
**Output** : posterior distribution $\{\mathcal{N}(\mu_i, \Sigma_i) \mid i = 1, \ldots, \mathcal{K}\}$

1 **Function** SVI $(\mathbf{x}_i, \vec{\mu}, \vec{\Sigma})$:
2  $\quad$ $\mu_i \leftarrow \vec{0}, \Sigma_i \leftarrow \mathbb{I}$ $\qquad$ // initialize with identity matrix
3  $\quad$ **while** *not converge* **do**
4  $\quad\quad$ $\mu_i, \Sigma_i \leftarrow (\mu_i, \Sigma_i) + \alpha * \nabla \mathcal{L}_{elbo}(\mathbf{x}_i; \mu_i, \Sigma_i)$
5  $\quad\quad$ // early stopping
6  $\quad\quad$ **if** $\vec{\Sigma} \neq []$ *and* $det(\Sigma_i) \geq avg(det(\vec{\Sigma}))$ **then**
7  $\quad\quad\quad$ break
8  $\quad\quad$ **end if**
9  $\quad$ **end while**
10 $\quad$ **return** $\mu_i, \Sigma_i$
11 $\vec{\mu} \leftarrow [], \vec{\Sigma} \leftarrow []$
12 /* offline session $\mathcal{S}_0$, perform supervised learning  */
13 // fine-tuning
14 **for** $\{(\mathbf{x}_i, y_i)\} \leftarrow \mathcal{D}_l$ **do**
15 $\quad$ $h(\cdot) \leftarrow h(\cdot) + \alpha * \nabla[(1 - \lambda)\mathcal{L}_{ce}(\mathbf{x}_i, y_i) + \lambda \mathcal{L}_{con}(\mathbf{x}_i)]$
16 **end for**
17 **for** $\{(\mathbf{x}_i, y_i)\} \leftarrow \mathcal{D}_l$ **do**
18 $\quad$ $\mu_i, \Sigma_i \leftarrow$ SVI$(h(\mathbf{x}_i), \vec{\mu}, \vec{\Sigma})$
19 $\quad$ $\vec{\mu}[y_i] \leftarrow \mu_i, \vec{\Sigma}[y_i] \leftarrow \Sigma_i$
20 **end for**
21 /* online session $\mathcal{S}_t$, perform unsupervised learning  */
22 $\mathcal{C}_l \leftarrow max(y_i)$
23 $\mathcal{C}_o \leftarrow \mathcal{C}_l$
24 **for** $\mathcal{S}_t \leftarrow \mathcal{S}_1$ **to** $\mathcal{S}_T$ **do**
25 $\quad$ **for** $\{\mathbf{x}_i\} \leftarrow \mathcal{D}_u^t$ **do**
26 $\quad\quad$ $\hat{y}_i \leftarrow clustering(h(\mathbf{x}_i); \mathcal{C}_o, \mathcal{C}_t)$
27 $\quad$ **end for**
28 $\quad$ // re-labeling
29 $\quad$ $\vec{\mu}' \leftarrow [], \vec{\Sigma}' \leftarrow []$
30 $\quad$ **for** $\{\mathbf{x}_i, \hat{y}_i\} \leftarrow (\mathcal{D}_u^t, \hat{Y})$ **do**
31 $\quad\quad$ $\mu_i', \Sigma_i' \leftarrow$ SVI$(h(\mathbf{x}_i), \vec{\mu}, \vec{\Sigma})$
32 $\quad\quad$ $\vec{\mu}'[\hat{y}_i] \leftarrow \mu_i', \vec{\Sigma}'[\hat{y}_i] \leftarrow \Sigma_i'$
33 $\quad$ **end for**
34 $\quad$ **for** $\{\mathbf{x}_i\} \leftarrow \mathcal{D}_u^t$ **do**
35 $\quad\quad$ $\bar{y}_i \leftarrow classifier(h(\mathbf{x}_i); \vec{\mu} \cup \vec{\mu}', \vec{\Sigma} \cup \vec{\Sigma}')$
36 $\quad$ **end for**
37 $\quad$ **for** $\{\mathbf{x}_i, \bar{y}_i\} \leftarrow (\mathcal{D}_u^t, \bar{Y})$ **do**
38 $\quad\quad$ $\mu_i, \Sigma_i \leftarrow$ SVI$(h(\mathbf{x}_i), \vec{\mu}, \vec{\Sigma})$
39 $\quad\quad$ $\vec{\mu}[\bar{y}_i] \leftarrow \mu_i, \vec{\Sigma}[\bar{y}_i] \leftarrow \Sigma_i$
40 $\quad$ **end for**
41 $\quad$ $\mathcal{C}_o \leftarrow \mathcal{C}_o + \mathcal{C}_t$
42 **end for**
43 **return** $\vec{\mu}, \vec{\Sigma}$

---

C-GCD differs from generalized category discovery (GCD) and novel category discovery (NCD) in the following two aspects:

- Unlabeled data is divided into multiple distinct online stages, where each stage $\mathcal{S}_t$ cannot access data from previous stages $\{\mathcal{S}_0, \mathcal{S}_1, \ldots, \mathcal{S}_{t-1}\}$. This means that the model can only use data from the current stage for learning.
- Unlabeled data and test sets for each online stage contain data from all previously encountered categories. This poses a challenge for the model, as it needs to distinguish between new and known categories while learning from the unlabeled data.

The objective of C-GCD is to improve the classification accuracy for all categories. This means that the model should strive to identify and learn to classify new categories in each stage without compromising the accuracy of previously learned categories (Feng et al., 2022; Li et al., 2023; Nguyen et al., 2018). The overall pseudocode for the proposed algorithm is shown in Algorithm 1.

## B. Learning from Unlabeled Data with Variational Inference

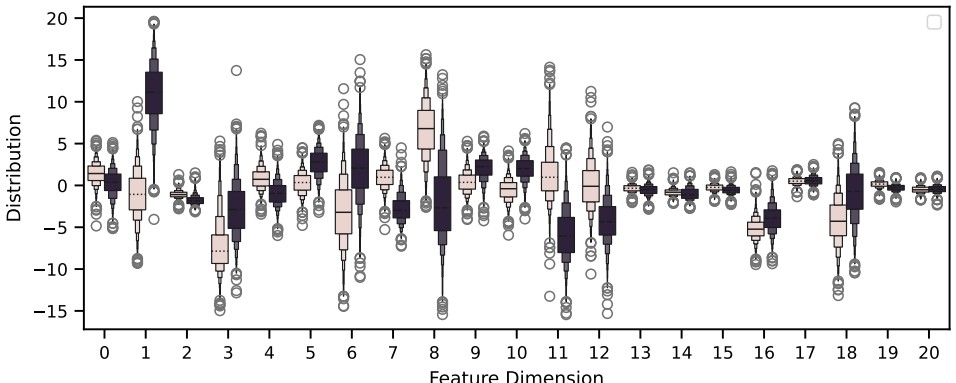

*Figure 4.* Comparison of the feature distributions for 8 dimensions across two different classes. A longer distribution indicates a higher variance of the dimension.

As illustrated in the Figure 4, the variances of data features across different dimensions exhibit significant differences, reflecting their varying contributions to classification. Modeling these features with MVN effectively captures such characteristic information.

*Table 7.* Accuracy comparison of various classification methods, including class mean (CM), point estimation (PE), and variational inference (VI).

| Datasets | CM | PE | VI |
|---|---|---|---|
| CIFAR-100 | 75.17 | 77.16 | 82.68 |
| ImageNet-100 | 86.64 | 88.0 | 90.18 |
| TinyImageNet | 71.47 | 70.66 | 76.95 |
| CUB-200 | 64.60 | 66.87 | 73.02 |

Variational inference (VI) optimizes the ELBO, effectively capturing feature distributions while mitigating the impact of outliers. We compared class modeling approaches based on statistical mean (CM), statistical distribution (PE), and variational inference. As shown in Table 7 and Figure 5, VI achieves superior classification accuracy, highlighting its effectiveness over traditional statistical methods.

## C. Experiment Details

### C.1. Dataset Details

We performed experiments on four datasets: CIFAR-100, ImageNet-100, TinyImageNet, and CUB-200. The specific configurations for each dataset are outlined in Table 8.

For comparison purposes, we conducted supervised learning on all datasets to determine the theoretical upper bound (Table 9) of model accuracy at the final session. Comparing this upper bound allows us to assess the performance and error

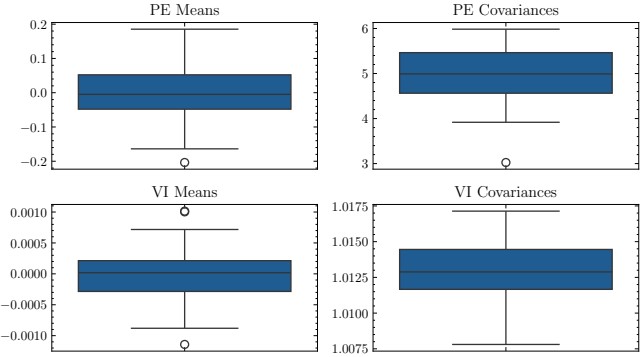

*Figure 5.* A comparison of the mean and covariance derived from PE and VI. It shows that the covariance in PE is an order of magnitude larger than that in VI. This indicates that many features in PE are scaled down to a much smaller range.

*Table 8.* The data configuration for the scenario with 50% categories labeled and 5 continual sessions.

| Datasets | Labeled | | Unlabeled | | |
|---|---|---|---|---|---|
| | categories | samples per category | novel categories | samples per novel | samples per known |
| CIFAR-100 | 50 | 400 | 10 | 400 | 25 |
| ImageNet-100 | 50 | 1000 | 10 | 1000 | 60 |
| TinyImageNet | 100 | 400 | 20 | 400 | 25 |
| CUB-200 | 100 | 25 | 20 | 25 | 5 |

margin of C-GCD methods.

*Table 9.* Accuracy upperbound of supervised learning.

| Datasets | Pre-Trained | Fine-Tuned |
|---|---|---|
| CIFAR-100 | 82.68 | 85.72 |
| ImageNet-100 | 90.18 | 90.24 |
| TinyImageNet | 76.95 | 81.14 |
| CUB-200 | 73.02 | 80.49 |

We further analyzed the upper bound of forgetting in a supervised learning manner. As demonstrated in Table 10, in a supervised learning scenario, the accuracy of a model trained on the complete dataset ('All($\mathcal{S}_0$)') is noticeably lower on the old class test set compared to a model trained exclusively on the old class data ('$\mathcal{S}_0$'). This decline is an inherent phenomenon, which can be termed as "inevitable forgetting".')')

## C.2. Metrics Details

*Accuracy*: The primary metric used to assess the classification performance of the models. The accuracy metrics are categorized as follows:

- "All": Accuracy on the test set that includes all categories encountered up to the current session $\mathcal{S}_t$, denoted as $ACC_{all} = \frac{1}{\|\mathcal{D}_{1:k+m}\|} \sum_{i=1}^{\|\mathcal{D}_{1:k+m}\|} \mathbb{1}(y_i = \tilde{y}_i \triangleq \mathcal{F}(\mathbf{x}_i; \Theta^t)); y_i \in \{1, \ldots, \mathcal{C}_{k+m}\}$.
- "Old": Accuracy on the test set comprising only the previously encountered classes, excluding the new classes introduced in the current session $\mathcal{S}_t$, denoted as $ACC_{old} = \frac{1}{\|\mathcal{D}_{1:k-1}\|} \sum_{i=1}^{\|\mathcal{D}_{1:k-1}\|} \mathbb{1}(y_i = \tilde{y}_i \triangleq \mathcal{F}(\mathbf{x}_i; \Theta^t)); y_i \in \{1, \ldots, \mathcal{C}_{k-1}\}$.
- "New": Accuracy on the test set of the newly introduced categories specific to the current session, denoted as $ACC_{new} = \frac{1}{\|\mathcal{D}_{k:k+m}\|} \sum_{i=1}^{\|\mathcal{D}_{k:k+m}\|} \mathbb{1}(y_i = \tilde{y}_i \triangleq \mathcal{F}(\mathbf{x}_i; \Theta^t)); y_i \in \{k, \ldots, \mathcal{C}_{k+m}\}$.

*Forgetting Rate*: After completing all online learning phases, the forgetting rate is defined as the difference between the final model's accuracy on the initially labeled dataset ($\mathcal{D}_l$) and the accuracy obtained during the initial offline supervised learning phase. This measures the degree of forgetting on the labeled dataset during the update process from $\mathcal{S}_0$ to $\mathcal{S}_T$, denoted as:

*Table 10.* Comparison of the accuracy on the $\mathcal{S}_0$ test set between models trained exclusively on $\mathcal{S}_0$ data and models trained on the entire dataset.

| Datasets | $\mathcal{S}_0$ | All($\mathcal{S}_0$) |
|---|---|---|
| CIFAR-100 | 88.90 | 82.68 |
| ImageNet-100 | 93.16 | 89.68 |
| TinyImageNet | 84.26 | 79.21 |
| CUB-200 | 80.43 | 73.54 |

$$\mathcal{M}_f = \frac{1}{\|\mathcal{D}_l\|} \sum_{i=1}^{\|\mathcal{D}_l\|} [\mathbb{1}(y_i = \tilde{y}_i \triangleq \mathcal{F}(\mathbf{x}_i; \Theta^0)) - \mathbb{1}(y_i = \tilde{y}_i \triangleq \mathcal{F}(\mathbf{x}_i; \Theta^T))]; y_i \in \{k, \dots, \mathcal{C}_l\}$$

*Novelty Learning*: The average learning performance of the new classes from $\mathcal{S}_1$ to $\mathcal{S}_T$, is evaluated by calculating the mean accuracy of the new classes across all sessions, denoted as: $\mathcal{M}_d = \frac{1}{T} \sum_{i=1}^{T} ACC_{new}^i$.

## C.3. Hyperparameters and Hardware.

The proposed method was implemented using Pytorch (Paszke et al., 2017) and JAX (Bradbury et al., 2018). For the feature extractor, we used a DINO (Caron et al., 2021) pre-trained ViT-B/16 model with weights from TorchHub. Fine-tuning was conducted via the BOFT algorithm (Liu et al., 2024), updating only the MLP and output dense layers of the ViT. The model was optimized using AdamW optimizer with a learning rate of $1e-3$ for 10 epochs.

To balance overall performance and runtime efficiency, we applied PCA to reduce the dimensionality of the output features to 384 dimensions. We employed the standard k-means algorithm for clustering, maintaining consistency with other baselines. For classifier training, a fixed learning rate of $1e-5$ was applied across all stages, with 1000 update steps conducted. All experiments were executed on an RTX 8000 GPU with 48GB of memory. All random seeds are set to 0.

## C.4. Impact of Hyperparameters

We conducted a hyperparameter $\mathcal{R}$ sensitivity analysis on C100 to evaluate the impact of early stopping on the performance of our method. The results are presented in Table 11 - 13.

*Table 11.* Comparisons in overall accuracy as $\mathcal{R}$ varies.

| $\mathcal{R}$ | $\mathcal{S}_0$ | $\mathcal{S}_1$ | $\mathcal{S}_2$ | $\mathcal{S}_3$ | $\mathcal{S}_4$ |
|---|---|---|---|---|---|
| 3 | 85.55 | 82.20 | 79.38 | 77.74 | 75.61 |
| 2 | 86.85 | 83.30 | 80.20 | 78.41 | 76.13 |
| 1 | 88.40 | 85.90 | 83.41 | 82.57 | 81.11 |
| 0 | **88.40** | **86.0** | **83.61** | **82.72** | **81.23** |
| -1 | 87.03 | 83.75 | 81.31 | 80.18 | 77.91 |
| -2 | 87.01 | 83.71 | 81.23 | 79.94 | 77.71 |

*Table 12.* Comparisons in the accuracy of old classes as $\mathcal{R}$ varies.

| $\mathcal{R}$ | $\mathcal{S}_0$ | $\mathcal{S}_1$ | $\mathcal{S}_2$ | $\mathcal{S}_3$ | $\mathcal{S}_4$ |
|---|---|---|---|---|---|
| 3 | 91.62 | 85.31 | 81.98 | 79.17 | 77.44 |
| 2 | 91.32 | 86.63 | 83.15 | 80.01 | 78.11 |
| 1 | 90.18 | 87.20 | 84.75 | 82.81 | 81.92 |
| 0 | **89.46** | **87.06** | **84.75** | **82.85** | **81.84** |
| -1 | 87.34 | 84.20 | 81.91 | 79.57 | 77.91 |
| -2 | 87.30 | 84.10 | 81.82 | 79.31 | 77.56 |

The experimental results indicate that the parameter $\mathcal{R}$ serves to balance the accuracy between new and old classes, with the overall accuracy being the best when $\mathcal{R} = 0$.

*Table 13.* Comparisons in the accuracy of new classes as $\mathcal{R}$ varies.

| $\mathcal{R}$ | $\mathcal{S}_0$ | $\mathcal{S}_1$ | $\mathcal{S}_2$ | $\mathcal{S}_3$ | $\mathcal{S}_4$ |
|---|---|---|---|---|---|
| 3 | 55.20 | 63.50 | 61.20 | 66.30 | 59.10 |
| 2 | 64.50 | 63.30 | 59.50 | 65.60 | 58.30 |
| 1 | 79.50 | 78.10 | 74.80 | 80.70 | 73.80 |
| 0 | **83.10** | **79.60** | **75.60** | **81.70** | **75.70** |
| -1 | 85.50 | 81.10 | 77.10 | 85.10 | 77.90 |
| -2 | 85.60 | 81.40 | 77.10 | 85.0 | 79.0 |

## C.5. Ablation Study

The following Table 14 presents the performance breakdown for each session in the ablation experiment.

*Table 14.* Comparison of the ablation study results across multiple datasets.

| Datasets | Methods | $\mathcal{S}_0$ | $\mathcal{S}_1$ | | | $\mathcal{S}_2$ | | | $\mathcal{S}_3$ | | | $\mathcal{S}_4$ | | | $\mathcal{S}_5$ | | |
|---|---|---|---|---|---|---|---|---|---|---|---|---|---|---|---|---|---|
| | | All | All | Old | New | All | Old | New | All | Old | New | All | Old | New | All | Old | New |
| C100 | w/o SVI[†] | 88.92 | 78.65 | 79.22 | 75.80 | 75.44 | 76.31 | 70.20 | 72.30 | 72.80 | 68.80 | 71.33 | 70.62 | 77.0 | 68.90 | 68.84 | 69.40 |
| | w/o fine-tuning | 88.54 | 82.51 | 85.52 | 67.5 | 79.25 | 80.81 | 69.90 | 76.71 | 78.35 | 65.20 | 75.13 | 76.01 | 68.10 | 73.48 | 73.94 | 69.30 |
| | w/o earlystop | 91.68 | 87.02 | 87.30 | **85.60** | 83.71 | 84.10 | **81.40** | 81.24 | 81.83 | 77.10 | 79.94 | 79.31 | **85.0** | 77.71 | 77.57 | **79.0** |
| | w/o re-labeling | 91.68 | 88.30 | 89.32 | 83.20 | 85.63 | 86.40 | 81.0 | 83.16 | 83.89 | **78.10** | 82.18 | 82.04 | 83.30 | 80.20 | 80.43 | 78.10 |
| | w/ all | **91.68** | **88.40** | **89.46** | 83.10 | **86.0** | **87.06** | 79.60 | **83.61** | **84.75** | 75.60 | **82.72** | **82.85** | 81.70 | **81.23** | **81.84** | 75.70 |
| Tiny | w/o SVI[†] | 84.78 | 75.03 | 77.26 | 63.90 | 70.47 | 72.31 | 59.40 | 66.86 | 68.72 | 53.80 | 63.74 | 65.30 | 51.30 | 61.14 | 61.91 | 54.20 |
| | w/o fine-tuning | 83.94 | 80.78 | 81.86 | 75.40 | 77.62 | 79.85 | 64.30 | 74.22 | 76.61 | 57.50 | 71.15 | 73.51 | 52.30 | 68.69 | 70.36 | 53.60 |
| | w/o earlystop | 88.32 | 83.36 | 83.96 | **80.40** | 79.91 | 80.55 | **76.10** | 76.82 | 78.15 | 67.50 | 73.76 | 74.95 | **64.30** | 71.23 | 71.95 | 64.70 |
| | w/o re-labeling | 88.32 | 85.03 | 86.22 | 79.10 | 82.04 | 83.13 | 75.50 | 78.89 | 80.47 | **67.80** | 76.11 | 77.59 | 64.30 | 73.67 | 74.61 | **65.20** |
| | w/ all | **88.32** | **85.10** | **86.44** | 78.40 | **82.40** | **84.15** | 71.90 | **79.35** | **81.38** | 65.10 | **76.72** | **78.43** | 63.0 | **74.52** | **75.88** | 62.20 |
| IN100 | w/o SVI[†] | 92.56 | 89.23 | 88.88 | 91.0 | 86.14 | 87.63 | 77.20 | 84.57 | 85.05 | 81.20 | 82.31 | 83.20 | 75.20 | 79.90 | 80.22 | 77.0 |
| | w/o fine-tuning | 93.04 | 92.10 | 91.36 | **95.80** | **90.80** | 90.40 | **93.20** | 88.47 | 89.77 | 79.40 | 86.53 | 87.17 | 81.40 | 84.0 | 85.60 | 69.60 |
| | w/o earlystop | 94.32 | 91.18 | 91.08 | 95.60 | 88.31 | 88.76 | 85.60 | 88.27 | 87.48 | **93.80** | 86.80 | 86.65 | **88.0** | 84.96 | 85.26 | 82.20 |
| | w/o re-labeling | 94.32 | 93.10 | 92.96 | 93.80 | 90.23 | 91.23 | 84.20 | 89.90 | 89.46 | 93.0 | 88.38 | 88.50 | 87.4 | 86.54 | 86.93 | **83.0** |
| | w/ all | **94.32** | **93.13** | **93.16** | 93.0 | 90.45 | **91.43** | 84.60 | **90.10** | **90.0** | 90.8 | **88.97** | **89.30** | 86.40 | **87.32** | **88.08** | 80.40 |
| CUB | w/o SVI[†] | 55.34 | 28.16 | 23.01 | 54.75 | 24.77 | 22.53 | 38.16 | 27.27 | 22.30 | 62.76 | 26.76 | 24.84 | 42.42 | 27.06 | 24.73 | 48.42 |
| | w/o fine-tuning | 80.16 | 68.77 | 74.18 | 40.84 | 61.43 | 65.88 | 34.92 | 57.81 | 58.82 | 55.59 | 53.50 | 56.12 | 32.40 | 51.10 | 52.18 | 41.22 |
| | w/o earlystop | 85.72 | 68.91 | 70.46 | **60.91** | 63.46 | 63.54 | **63.03** | 61.31 | 60.57 | **66.60** | 58.11 | 58.61 | **54.04** | 56.43 | 56.12 | **59.29** |
| | w/o re-labeling | 85.72 | 77.95 | 81.94 | 57.39 | 72.97 | 74.72 | 62.52 | 69.82 | 70.32 | 66.26 | 65.91 | 67.46 | 53.17 | 63.32 | 63.80 | 58.95 |
| | w/ all | **85.72** | **78.46** | **82.92** | 55.45 | **74.60** | **76.72** | 62.01 | **71.84** | **73.38** | 60.83 | **68.60** | **70.91** | 49.64 | **66.44** | **67.76** | 54.38 |

We also conducted an ablation study on the benefit of introducing Mahalanobis distance, and the results obtained by replacing the Mahalanobis with the Euclidean distance are shown in Table 15.

*Table 15.* Ablation study results without Mahalanobis.

| Datasets | Methods | All($\mathcal{S}_0$)$\uparrow$ | **All($\mathcal{S}_5$)$\uparrow$** | $\mathcal{M}_f \downarrow$ | $\mathcal{M}_d \uparrow$ |
|---|---|---|---|---|---|
| C100 | w/o Mahalanobis | 90.52 | 76.06 | 10.10 | 74.96 |
| Tiny | w/o Mahalanobis | 87.64 | 70.33 | 9.70 | 67.50 |
| IN100 | w/o Mahalanobis | 93.60 | 85.06 | 8.04 | 86.56 |
| CUB | w/o Mahalanobis | 83.78 | 55.79 | 0.82 | 28.15 |

Compared to VB-CGCD with Mahalanobis distance, the overall final accuracy decreased by approximately 5.56 on average. This is because Euclidean distance, being a special case of Mahalanobis, is susceptible to collapse in high-dimensional spaces. By incorporating covariance information, Mahalanobis distance effectively mitigates this issue, thereby achieving a more optimal classification boundary.

## C.6. Parameter Size

In the proposed method, the primary storage demand arises from the covariance matrix, whose size scales quadratically with the number of feature dimensions. However, leveraging the symmetric property of covariance matrices, only the lower triangular part needs to be stored, effectively halving the storage requirements.

To examine the sensitivity of the proposed method to the structure of the covariance matrix, we also designed a simplified model that uses a diagonal covariance matrix. As shown in Table 17, while there is a noticeable performance gap, the diagonal matrix model still outperforms other baselines. Furthermore, dimensionality reduction techniques, such as PCA, can be applied to reduce the number of feature dimensions, thereby dramatically decreasing the size of the covariance matrix while maintaining competitive performance.

*Table 16.* Comparison of performance with varying parameter sizes.

| Datasets | Methods | $\mathcal{S}_0$ All | $\mathcal{S}_1$ All | Old | New | $\mathcal{S}_2$ All | Old | New | $\mathcal{S}_3$ All | Old | New | $\mathcal{S}_4$ All | Old | New | $\mathcal{S}_5$ All | Old | New |
|---|---|---|---|---|---|---|---|---|---|---|---|---|---|---|---|---|---|
| C100 | diagonal | 90.46 | 83.78 | 83.46 | **85.40** | 79.59 | 79.95 | 77.40 | 76.70 | 77.74 | 69.40 | 75.88 | 75.64 | 77.80 | 74.11 | 75.49 | 61.70 |
|  | dim=100 | 90.86 | 86.48 | 87.08 | 83.50 | 83.02 | 84.10 | 77.80 | 80.09 | 81.29 | 71.70 | 78.82 | 78.45 | 81.80 | 76.41 | 76.59 | 74.80 |
|  | dim=200 | 91.34 | 86.86 | 87.28 | 84.80 | 83.80 | 84.35 | **80.50** | 81.12 | 81.92 | 75.50 | 79.94 | 79.41 | **84.20** | 77.73 | 77.75 | **77.50** |
|  | dim=384 | **91.68** | **88.40** | **89.46** | 83.10 | **86.0** | **87.06** | 79.60 | **83.61** | **84.75** | **75.60** | **82.72** | **82.85** | 81.70 | **81.23** | **81.84** | 75.70 |
| Tiny | diagonal | 86.66 | 82.20 | 82.90 | 78.70 | 79.56 | 81.05 | 70.60 | 76.19 | 79.03 | 56.30 | 73.03 | 75.97 | 49.50 | 70.04 | 72.74 | 45.70 |
|  | dim=100 | 87.40 | 82.90 | 83.90 | 77.90 | 79.20 | 80.27 | 72.80 | 75.96 | 77.56 | 64.80 | 72.66 | 74.11 | 61.0 | 69.94 | 70.97 | 60.70 |
|  | dim=200 | 88.06 | 83.75 | 84.52 | **79.90** | 80.48 | 81.18 | **76.30** | 77.42 | 78.85 | **67.40** | 74.32 | 75.68 | **63.40** | 71.87 | 72.75 | **63.90** |
|  | dim=384 | **88.32** | **85.10** | **86.44** | 78.40 | **82.40** | **84.15** | 71.90 | **79.35** | **81.38** | 65.10 | **76.72** | **78.43** | 63.0 | **74.52** | **75.88** | 62.20 |
| IN100 | diagonal | 93.20 | 84.97 | 82.36 | **98.0** | 79.63 | 78.90 | 84.0 | 80.25 | 78.80 | 90.40 | 79.44 | 78.88 | 84.0 | 78.02 | 78.91 | 70.0 |
|  | dim=100 | 93.72 | 91.47 | 90.88 | 94.40 | 88.20 | 89.0 | 83.40 | 87.57 | 87.06 | 91.20 | 86.04 | 86.13 | 85.40 | 84.26 | 84.42 | **82.80** |
|  | dim=200 | 93.92 | 92.0 | 91.40 | 95.0 | 88.82 | 89.46 | **85.20** | 88.37 | 87.82 | **92.20** | 86.93 | 86.87 | **87.40** | 85.16 | 85.44 | 82.60 |
|  | dim=384 | **94.32** | **93.13** | **93.16** | 93.0 | **90.45** | **91.43** | 84.60 | **90.10** | **90.0** | 90.8 | **88.97** | **89.30** | 86.40 | **87.32** | **88.08** | 80.40 |

*Table 17.* Comparison of performance with varying parameter sizes.

| Datasets | Methods | ACC($\mathcal{S}_0$)↑ | All($\mathcal{S}_5$)↑ | $\mathcal{M}_f$ ↓ | $\mathcal{M}_d$ ↑ |
|---|---|---|---|---|---|
| C100 | diagonal | 90.46 | 74.11 | 15.26 | 74.34 |
|  | dim=100 | 90.86 | 76.41 | 13.62 | 77.92 |
|  | dim=384 | 91.68 | 81.23 | 6.94 | 79.14 |
| Tiny | diagonal | 86.66 | 70.04 | 5.86 | 60.16 |
|  | dim=100 | 87.40 | 69.94 | 12.28 | 67.44 |
|  | dim=384 | 88.32 | 74.52 | 5.72 | 68.12 |
| IN100 | diagonal | 93.20 | 78.02 | 21.48 | 85.28 |
|  | dim=100 | 93.72 | 84.26 | 10.56 | 87.44 |
|  | dim=384 | 94.32 | 87.32 | 5.68 | 87.04 |

## C.7. Experiments in Diverse Scenarios

To explore the practical applicability of the proposed method, we designed two sets of experiments. The first is an extended experiment based on half-classes labeled, where the number of online sessions is expanded from 5 (denoted as 'b50t5') to 10 (referred to as 'b50t10'). The experimental setup is as follows:

*Table 18.* The data configuration for the scenario with 50% categories labeled and 10 continual sessions.

| Datasets | Labeled categories | samples per category | Unlabeled novel categories | samples per novel | samples per known |
|---|---|---|---|---|---|
| CIFAR-100 | 50 | 400 | 5 | 400 | 25 |
| ImageNet-100 | 50 | 1000 | 5 | 1000 | 60 |
| TinyImageNet | 100 | 400 | 10 | 400 | 25 |

Compared to the 5 online sessions experiment, this setup reduces the difficulty of the clustering algorithm, thereby

diminishing some errors caused by clustering inherent. However, it simultaneously introduces more noise data from known classes (nearly double). As shown in the results table, the performance decline compared to b50t5 is only about 1–3%, demonstrating the robustness and noise resistance of the proposed algorithm.

*Table 19.* Performance of each session with the b50t10 setting.

| Datasets | $\mathcal{S}_0$ | $\mathcal{S}_1$ | $\mathcal{S}_2$ | $\mathcal{S}_3$ | $\mathcal{S}_4$ | $\mathcal{S}_5$ | $\mathcal{S}_6$ | $\mathcal{S}_7$ | $\mathcal{S}_8$ | $\mathcal{S}_9$ | $\mathcal{S}_{10}$ |
|---|---|---|---|---|---|---|---|---|---|---|---|
| C100 | 91.68 | 90.58 | 88.51 | 86.56 | 85.30 | 83.90 | 83.22 | 82.05 | 80.40 | 80.24 | 79.28 |
| Tiny | 88.32 | 85.80 | 84.16 | 82.16 | 80.91 | 78.88 | 78.27 | 76.85 | 75.30 | 74.35 | 73.45 |
| IN100 | 94.32 | 93.96 | 91.36 | 88.46 | 87.45 | 87.04 | 86.40 | 85.50 | 85.20 | 85.07 | 84.48 |

*Table 20.* Performance of each session with the b10t9 setting.

| Data | Methods | $\mathcal{S}_0$ | $\mathcal{S}_1$ | $\mathcal{S}_2$ | $\mathcal{S}_3$ | $\mathcal{S}_4$ | $\mathcal{S}_5$ | $\mathcal{S}_6$ | $\mathcal{S}_7$ | $\mathcal{S}_8$ | $\mathcal{S}_9$ | b50-All($\mathcal{S}_5$) |
|---|---|---|---|---|---|---|---|---|---|---|---|---|
| C100 | VB-CGCD | 98.90 | 93.55 | 86.60 | 82.95 | 81.22 | 79.31 | 78.02 | 76.50 | 76.31 | 74.72 | 81.32 |
| TINY | VB-CGCD | 92.70 | 81.05 | 79.56 | 76.05 | 75.0 | 73.75 | 71.52 | 70.26 | 68.20 | 66.30 | 74.52 |
| IN100 | VB-CGCD | 98.20 | 96.20 | 91.26 | 91.35 | 90.08 | 89.76 | 87.88 | 85.80 | 83.62 | 82.88 | 87.32 |

The second experiment evaluates the proposed method in an extremely limited labeled categories scenario. In this setup, only 10% of the classes are labeled, while the remaining 90% are unlabeled and evenly distributed across 9 subsequent online sessions, denoted as 'b10t9'. As shown in Table 20, the proposed method demonstrates strong performance, maintaining high accuracy even with significantly reduced labeled data. These results underscore the robustness and adaptability of the method in scenarios with scarce labeled data.

# D. Extensions

## D.1. Handling Unknown Number of Novel Classes

In our prior experiments, the number of unknown categories per session was fixed. For scenarios where the count of new classes is unknown, we note that numerous off-the-shelf estimation methods (Ronen et al., 2022; Cendra et al., 2024) exist, all of which can be seamlessly integrated into VB-CGCD. For demonstration purposes, we employed the classic silhouette score to implement a method for estimating the number of unknown classes, which we integrated into VB-CGCD. The results are shown in Table 21.

*Table 21.* VB-CGCD's performance when the number of new categories is unknown.

| Datasets | $\mathcal{S}_0$ | $\mathcal{S}_1$ | | | $\mathcal{S}_2$ | | | $\mathcal{S}_3$ | | | $\mathcal{S}_4$ | | | $\mathcal{S}_5$ | | |
|---|---|---|---|---|---|---|---|---|---|---|---|---|---|---|---|---|
| | All | All | Old | New | All | Old | New | All | Old | New | All | Old | New | All | Old | New |
| C100 | 91.68 | 82.48 | 87.20 | 58.90 | 79.08 | 82.31 | 59.70 | 76.25 | 78.90 | 57.70 | 75.24 | 76.07 | 68.60 | 73.13 | 74.94 | 56.80 |
| Tiny | 88.32 | 85.55 | 86.82 | 79.20 | 82.41 | 84.01 | 72.80 | 79.87 | 81.41 | 69.10 | 77.20 | 78.82 | 64.20 | 74.77 | 76.34 | 60.60 |
| IN100 | 94.32 | 91.93 | 93.12 | 86.0 | 89.45 | 90.43 | 83.60 | 88.60 | 88.28 | 90.80 | 86.11 | 87.62 | 74.0 | 84.56 | 85.08 | 79.80 |
| CUB | 85.72 | 79.98 | 83.68 | 60.91 | 65.98 | 67.42 | 57.41 | 63.91 | 65.94 | 49.47 | 57.54 | 58.24 | 51.76 | 54.40 | 57.54 | 25.61 |

Due to errors in estimating the number of unknown classes, the clustering error is exacerbated, ultimately leading to an average overall accuracy reduction of approximately 5.66. Nonetheless, it demonstrates the scalability of our method to handle scenarios with an unknown number of types, and our performance still outperforms the SOTA method, Happy, which also uses the silhouette score, as shown in Table 22.

## D.2. VB-CGCD as a Replay Mechanism

The proposed method, when combined with the NCM classifier, can serve as a standalone C-GCD algorithm. Additionally, VB-CGCD can be utilized for distribution modeling in other C-GCD approaches that rely on Gaussian distributions as a replay or prompt mechanism, enabling seamless integration with these methods.

*Table 22.* The average accuracy across 5 sessions on C100 with silhouette score.

| Methods | All | Old | New |
|---------|-----|-----|-----|
| Happy | 68.80 | 72.40 | 45.74 |
| VB-CGCD | 77.23 | 79.88 | 60.34 |

To demonstrate its versatility, we designed an experiment where the posterior distributions obtained by VB-CGCD were used as a replay mechanism to train a neural network classifier. In this experiment, the neural network does not learn directly from the training data but instead from samples drawn from the Gaussian distributions generated by VB-CGCD. During sampling, an equal number of samples is drawn uniformly for each class, mitigating label bias and avoiding catastrophic forgetting, as the neural network is retrained with a complete and balanced set of data in each session. Experimental results (see Table 23) show that, despite a roughly $3\%$ decrease in accuracy, the achieved accuracy is almost identical to that of VB-CGCD, the MLP classification head derived from this process still surpasses existing baselines. This demonstrates the potential of the method as a promising extension for broader applications.

*Table 23.* The accuracy of training an MLP when using VB-CGCD as the replay mechanism (dim=100).

| Datasets | Methods | $\mathcal{S}_0$ | $\mathcal{S}_1$ | | | $\mathcal{S}_2$ | | | $\mathcal{S}_3$ | | | $\mathcal{S}_4$ | | | $\mathcal{S}_5$ | | |
|----------|---------|------|------|------|------|------|------|------|------|------|------|------|------|------|------|------|------|
| | | All | All | Old | New | All | Old | New | All | Old | New | All | Old | New | All | Old | New |
| C100 | VB-CGCD | 90.86 | 86.48 | 87.08 | 83.50 | 83.20 | 84.10 | 77.80 | 80.08 | 81.28 | 71.70 | 78.82 | 78.45 | 81.80 | 76.41 | 76.58 | 74.80 |
| | VB-CGCD+MLP | 90.68 | 85.36 | 87.0 | 77.20 | 81.41 | 82.56 | 74.50 | 77.98 | 79.68 | 66.10 | 76.33 | 76.08 | 78.30 | 73.36 | 73.40 | 73.30 |
| Tiny | VB-CGCD | 87.40 | 82.90 | 83.90 | 77.90 | 79.20 | 80.26 | 72.80 | 75.96 | 77.56 | 64.80 | 72.66 | 74.11 | 61.0 | 69.94 | 70.97 | 60.70 |
| | VB-CGCD+MLP | 87.21 | 80.78 | 81.42 | 77.60 | 77.14 | 78.0 | 72.0 | 72.46 | 73.76 | 63.40 | 68.83 | 70.28 | 57.20 | 65.95 | 66.79 | 58.40 |
| IN100 | VB-CGCD | 93.72 | 91.47 | 90.88 | 94.40 | 88.20 | 89.0 | 83.40 | 87.58 | 87.06 | 91.20 | 86.04 | 86.13 | 85.40 | 84.26 | 84.42 | 82.80 |
| | VB-CGCD+MLP | 93.11 | 90.67 | 90.24 | 92.80 | 87.20 | 88.53 | 79.20 | 86.53 | 85.86 | 91.20 | 84.02 | 83.95 | 84.60 | 82.0 | 82.22 | 80.0 |

## D.3. Upgrade to DINOv2

DINO recently introduced a new pre-trained version, DINOv2, demonstrating significant performance improvements. We upgraded the backbone network in VB-CGCD method to DINOv2-ViT-B/14, and the results are shown in Table 24. While all datasets showed slight improvement, the accuracy on ImageNet100 remained consistent. This may be attributed to the fact that this dataset was already well-leveraged during the pre-training phase of DINO and DINOv2, leaving little room for further gains.

*Table 24.* Performance with DINOv2

| Datasets | Methods | $\mathcal{S}_0$ | $\mathcal{S}_1$ | | | $\mathcal{S}_2$ | | | $\mathcal{S}_3$ | | | $\mathcal{S}_4$ | | | $\mathcal{S}_5$ | | |
|----------|---------|------|------|------|------|------|------|------|------|------|------|------|------|------|------|------|------|
| | | All | All | Old | New | All | Old | New | All | Old | New | All | Old | New | All | Old | New |
| C100 | VB-CGCD+DINOv2 | 94.88 | 91.41 | 92.0 | 88.50 | 88.78 | 89.51 | 84.40 | 86.42 | 87.02 | 82.20 | 84.46 | 85.38 | 77.10 | 84.03 | 83.28 | 90.70 |
| Tiny | VB-CGCD+DINOv2 | 92.10 | 88.48 | 88.74 | 87.20 | 85.22 | 86.38 | 78.30 | 82.28 | 83.57 | 73.30 | 79.64 | 81.03 | 68.50 | 76.87 | 78.12 | 65.60 |
| IN100 | VB-CGCD+DINOv2 | 96.44 | 93.16 | 96.08 | 78.60 | 92.45 | 93.0 | 89.20 | 90.42 | 92.31 | 77.20 | 88.60 | 90.25 | 75.40 | 87.28 | 88.53 | 76.0 |
| CUB | VB-CGCD+DINOv2 | 92.45 | 84.92 | 86.13 | 78.69 | 79.75 | 82.21 | 65.07 | 76.26 | 77.11 | 70.28 | 72.85 | 74.80 | 56.86 | 70.40 | 71.49 | 60.35 |

