# OpenReview forum: "Continual Generalized Category Discovery: Learning and Forgetting from a Bayesian Perspective"
_ICML.cc/2025/Conference — ICML 2025 poster_

### Official Review · Reviewer_uAHx · 2025-03-03

**Overall Recommendation:** 3

**Summary:**

This paper introduces Variational Bayes Continual Generalized Category Discovery (VB-CGCD), a Bayesian framework designed to address key challenges in Continual Generalized Category Discovery (C-GCD), including label bias, pseudo-label errors, and the learning-forgetting tradeoff. VB-CGCD comprises four key components: (1) self-supervised offline fine-tuning, (2) self-corrective re-labeling, (3) variational Bayesian distribution estimation with a nearest-class-mean classifier, and (4) covariance-driven early stopping. Experimental results demonstrate that VB-CGCD outperforms existing methods across standard and newly proposed benchmarks with minimal labeled data.

**Claims And Evidence:**

1. The problem formulation in Eq. (1) requires that the total losses on previous t-1 tasks with parameters leanred after task t-1, does not be increased after learning task t. How is this gaurantee ensured? Does this optimization objective always have a valid solution? It seems that VB-CGCD could not fulfill this constraint, as shown in Figure 2 (a), the accuracy of previous tasks decreases after learning new tasks.

2. The paper states: “To mitigate feature drift, we perform fine-tuning exclusively during the offline stages and keep the backbone network parameters frozen throughout all sessions, including offline and online classification learning.” However, freezing the backbone may hinder the model’s plasticity, particularly when significant distribution shifts occur between sessions. Further discussion on the trade-off between stability and adaptability would strengthen the paper.

3. One of the core ideas, i.e., modeling each class as a multivariate normal distribution (MVN), has been explored in previous continual learning works [1-4].

[1] Fecam: Exploiting the heterogeneity of class distributions in exemplar-free continual learning. NIPS2023

[2] Learning semi-supervised gaussian mixture models for generalized category discovery. ICCV 2023

[3] Steering prototype with prompt-tuning for rehearsal-free continual learning. WACV 2024

[4] Happy: A Debiased Learning Framework for Continual Generalized Category Discovery, NIPS 2024

**Essential References Not Discussed:**

Several continual learning works [5-9] that leverage variational inference should be discussed in the related work section for a more comprehensive comparison.

[5] Variational continual learning, ICLR 2018

[6] Variational auto-regressive gaussian processes for continual learning, ICML 2021

[7] Generalized variational continual learning, ICLR 2021

[8] Continual learning via sequential function-space variational inference, ICML 2022

[9] Continual variational autoencoder learning via online cooperative memorization, ECCV 2022

**Experimental Designs Or Analyses:**

Variational inference-based methods are generally computationally expensive. However, the paper does not compare the computational cost of VB-CGCD with other methods, focusing only on parameter sizes. A computational efficiency analysis would enhance the evaluation.

**Methods And Evaluation Criteria:**

Yes.

**Other Comments Or Suggestions:**

1. It is recommended to include punctuation in each numbered equation.

2. Please ensure the correct quotation marks are used throughout the paper.

**Other Strengths And Weaknesses:**

Other stengths.

The improvement over previous methods is substantial.

**Questions For Authors:**

1. When the number of new categories is unknown, how does VB-CGCD estimate it effectively?

2. Can you further explain how the class merging is conducted? Specifically, how are “the latent variables of old classes and the pseudo-labeled distribution merged into a unified MVN-NCM classifier”?

3. According to the ablation study results in Table 11, early stopping and re-labeling seem to hinder the learning of new classes. Does this suggest that the proposed method prioritizes stability (performance on old classes) over plasticity (performance on new classes) to achieve a tradeoff?

**Relation To Broader Scientific Literature:**

The key contributions of this paper aim to address the challenge of learning from both known and novel categories in an evolving data stream. By providing a practical solution, it aligns with broader efforts in the field to advance continual generalized category discovery.

**Theoretical Claims:**

The paper presents the theoretical upper bound (Table 9) of model accuracy at the final session, but does not provide formal proofs. This upper bound might come from experiments.

---

> ### Author Rebuttal · Authors · 2025-04-01
>
> ## Claims And Evidence:
>
> >1.
>
> The constrained optimization equation formally represents the objective of "learning new classes without forgetting old ones." We observed that maximizing overall accuracy inevitably leads to a decline in the classification performance of previous classes, as illustrated in P2 of Figure 2. Therefore, considering the interplay between old and new classes, we believe that this issue is a multi-objective optimization problem. Accordingly, we relax the original constraint by imposing a covariance constraint, which approximates a balance between preventing forgetting and effectively learning new classes.
>
> >2.
>
> With advancements in self-supervised pre-training, the effectiveness of self-supervised fine-tuning in small-scale scenarios has diminished. On the other hand, due to the presence of classification errors, employing supervised fine-tuning during online sessions can introduce a significant number of misclassifications, which in turn leads to a decline in overall performance. Meanwhile, we did not entirely freeze the backbone; instead, we performed fine-tuning with labeled data during the offline phase, which was performed using verified ground-truth labels.
>
> >3.
>
> Most methods use the MVN as a sampling distribution for data replay or to assist clustering, rather than directly as a classification prototype. In contrast, VB-CGCD approximates the MVN through variational inference, and we believe that this strategy is the key to our improved performance.
>
> ## Experimental Designs Or Analyses:
>
> We acknowledge that Variational Inference can be computationally intensive. However, our VI process does not need to reach full convergence; only a few steps (1000 epochs in our experiments) are sufficient to distinguish different categories. Additionally, by not requiring fine-tuning of the backbone network during training, we only need to extract features once for each sample, performing a single forward pass through the backbone, which significantly reduces training time. Consequently, compared to other methods that necessitate frequent fine-tuning of the backbone network, our training time is substantially reduced. The running time during training on one RTX-8000 is shown as follows:
>
> |Datasets|OfflineFine-Tuning (mins)|OfflineSession (mins)|OnlineSession (mins)|Overall (mins)|
> |-|-|-|-|-|
> |C100|67|3.03|1.51|97|
> |TINY|112|2.27|1.58|145|
> |IN100|140|3.36|1.22|169|
> |CUB|12.09|2.01|1.12|39|
>
> Compared to HAPPY (over 25 hours of training) and PromptCCD (over 40 hours of training), VB-CGCD demonstrates significantly superior training efficiency.
> After training, VB-CGCD achieves inference computational efficiency comparable to other prototype-based approaches, normally in milliseconds.
>
> ## Q&A
>
> >1.
>
> As a dynamically extensible incremental learning classifier, VB-CGCD can be integrated with advanced clustering algorithms, including those capable of dynamically estimating the number of categories. Since VB-CGCD does not require a preset number of categories, it can be combined with many off-the-shelf methods for estimating unknown category counts. We employed the classical silhouette score to implement a method for estimating the number of unknown categories and integrated it into VB-CGCD. For experimental results, please refer to Response 2 of Reviewer 1.
>
> >2.
>
> VB-CGCD models each category as an independent probability distribution and maintains a collection of all category distributions (i.e., a matrix of latent variables) in memory. After estimating the pseudo-label distribution, we incorporate it into the distribution collection (prototype matrix). During re-labeling, the distances between all samples and this collection are computed to assign labels accordingly. Subsequently, the pseudo-distribution is removed from the collection, and a new distribution is re-estimated. The merge operation is purely an array manipulation, which accounts for VB-CGCD’s inherent scalability, and the reason for being able to handle unknown number of categories.
>
> >3.
>
> Since the primary goal of continual learning is to enhance overall learning performance, we agree that preventing overfitting to new classes is necessary to mitigate the forgetting of previously acquired knowledge. In fact, many continual learning methods, whether by adding regularization constraints or employing data replay, aim to some extent to slow down the learning of new classes to prevent forgetting old ones. Without imposing any constraints and allowing unrestricted learning of new classes, the model's classification accuracy on new classes may continue to improve, but the accuracy on old classes could drop drastically, as illustrated in phase P3 of Figure 2. Therefore, to maximize overall accuracy, continual learning must strike a balance between stability (preventing forgetting) and plasticity (learning new categories). This trade-off inherently implies that optimizing one aspect will inevitably come at the expense of the other.

---

> > ### Comment · Reviewer_uAHx · 2025-04-04
> >
> > Thank’s for the rebuttal, which addresses some of my concerns. However, I still believe that the optimization objective in Eq. (1) may not be well-aligned with the proposed method. Introducing a relaxation to the constraint could be beneficial, yet this aspect appears to be missing in the original paper. In addition, while the authors provide an explanation regarding the effect of self-supervised pre-training, they do not directly address my concern about freezing the model after the offline stage, which may limit the model’s plasticity. Lastly, the absence of discussion on relevant works [1–9] weakens the paper’s comprehensiveness. Therefore, I would prefer to maintain my original recommendation at this stage.

---

> > > ### Author Response · Authors · 2025-04-06
> > >
> > > ## 1. Eq(1)
> > >
> > > Eq(1) is a general optimization objective for class incremental learning, which originates from the classical work on continual learning GEM [1]. In our discussion of the P2 phase in Figure 2, we argue that this objective can only achieve a suboptimal outcome (akin to the P1 phase). **Eq (1) is not directly related to our work and was not the focus of our optimization**; it was included merely as a general definition in the class incremental learning problem formulation.
> > >
> > > *We will avoid this ambiguity in the revised version.*
> > >
> > > >[1] Gradient Episodic Memory for Continual Learning. NeurIPS 2017.
> > >
> > > ## 2. Model plasticity
> > >
> > > We employ the pretrained model as a frozen backbone network for feature extraction, which encodes raw images into features. **The model's plasticity is principally derived from the Bayesian Neural Network (BNN)-based prototype classifier constructed atop this backbone.**
> > >
> > > (1) This BNN classifier parametrically models each category's distribution as **learnable multivariate Gaussian distributions**. The strong approximation capacity of multivariate Gaussians endows the classifier with plasticity for incremental learning.
> > >
> > > (2) The per-class independent distribution fitting mechanism enables **dynamic expansion**, allowing the model to adaptively accommodate arbitrary numbers of novel categories across domains.
> > >
> > > Notably, baselines like HAPPY and PromptCCD also adopt backbone freezing with only final-block tuning. **Their plasticity primarily stems from MLP classification heads**, analogous to our BNN's functional role. Compared to an MLP classification head, BNN offers more flexible dynamic expansion and enhanced adaptability. Besides, many SOTA CIL approaches freeze the backbone while solely training classifiers atop it [1]. This design philosophy preserves the strong generalization capabilities of pretrained models. Representative methods, including **FeCAM, L2P [2], and DualPrompt [3], strictly maintain the frozen pretrained backbone without any fine-tuning**.
> > >
> > > >[1] Class-Incremental Learning: A Survey. TPMI, 2024.
> > >
> > > >[2] Learning to Prompt for Continual Learning. CVPR 2022.
> > >
> > > >[3] DualPrompt: Complementary Prompting for Rehearsal-free Continual Learning, ICCV 2023.
> > >
> > > To demonstrate the plasticity of our model, we continued learning on IN100 after having completed incremental learning on C100 with the pretrained backbone and BNN classifier. The model sequentially learns IN100 categories while retaining C100 knowledge, thereby validating its adaptability.
> > > |S0|S1|||S2|||S3|||S4|||S5|||
> > > |-|-|-|-|-|-|-|-|-|-|-|-|-|-|-|-|
> > > |All|All|Old|New|All|Old|New|All|Old|New|All|Old|New|All|Old|New|
> > > |82.24|81.42|81.94|76.3|81.04|81.31|78.0|80.0|80.82|70.2|79.55|79.82|76.0|78.6|79.31|68.6|
> > >
> > > The result shows that VB-CGCD achieved a final accuracy of 78.6, demonstrating its ability to effectively classify both C100 and IN100, which underscores its strong cross-domain adaptability.
> > >
> > > ## 3. Missing Related Works
> > >
> > > GPC [2] employs a GMM for clustering and estimates unknown category numbers through split-merge mechanisms, focusing primarily on clustering rather than classification. FeCAM [1] utilizes Gaussian prototypes obtained via statistical methods, making them non-learnable and sensitive to data scale variations. In contrast, VB-CGCD employs a learnable BNN to establish Gaussian prototypes, offering greater generality and effectively handling covariance differences between classes.
> > >
> > > CPP [3] uses class means as prototypes (prompts) and integrates a Transformer as the classifier, whereas VB-CGCD employs Gaussian distributions along with a non-parametric distance function, enhancing robustness and interpretability. HAPPY [4] utilizes Gaussian prototypes as a replay mechanism to mitigate forgetting, while VB-CGCD is replay-free and directly utilizes prototypes for classification.
> > >
> > > VCL [5], GVCL [7], VAR-GP [6], and S-FSVI [8] are methods that approach continual learning as a sequence of tasks, utilizing variational inference to regularize parameter updates via the KL divergence. These techniques bridge Bayesian inference with continual learning, providing insights into the trade-off between plasticity and stability. These methods employ variational inference for regularization in learning likelihood probabilities, while VB-CGCD leverages variational inference to directly learn generative models of data distributions, enabling classification through distance functions. OCM [9] uses variational autoencoders to learn data distributions, serving as samplers for replay mechanisms to mitigate forgetting, whereas VB-CGCD operates without the need for replay.
> > >
> > > *The revised version will include these in the related work, clarifying technical distinctions and providing a comprehensive discussion.*
> > >
> > > We hope that our responses have addressed your concerns and would greatly appreciate it if you could consider raising our score. Also, let us know if there are any more concerns.

---

### Official Review · Reviewer_9UpM · 2025-03-11

**Overall Recommendation:** 3

**Summary:**

This paper addresses Continual Generalized Category Discovery (C-GCD), or say, iGCD, a task where a model must incrementally learn new classes from unlabeled data streams while preserving knowledge of previously learned classes, a challenge exacerbated by mixed-class data streams and catastrophic forgetting. C-GCD is a task that involves: IL, semi-supervised learning, and class discovery.

In this work, the authors Variational Bayes C-GCD (VB-CGCD), a Bayesian framework that uses variational inference to model class distributions, align covariances between old and new classes, and mitigate forgetting through a covariance-aware nearest-class-mean (NCM) classifier and an early stopping mechanism.

The proposed method, VV-CGCD is evaluated on well-established GCD benchmarks: CIFAR-100, TinyImageNet, ImageNet-100, and CUB-200. The results show improvements over the prev. arts.

**Claims And Evidence:**

Yes.

**Essential References Not Discussed:**

There is a significant lack of discussion on related work in this study.

First, the coverage of the Novel Class Discovery (NCD) and Generalized Category Discovery (GCD) literature is very limited, with only two works from each category being discussed. However, a broader spectrum of NCD techniques is highly relevant to this work, as they serve as the foundation for both GCD and Category-wise GCD (C-GCD). A more comprehensive discussion is needed to highlight the technical and methodological similarities between this study and existing NCD and GCD approaches.

Second, there is a notable omission of discussions and comparisons regarding Class-incremental NCD, which forms the basis of C-GCD. Foundational works in this domain [1, 2, 3] are neither reviewed nor compared, despite their relevance. Addressing these gaps would strengthen the contextualization of this work within the broader research landscape.

[1] Joseph K J, Paul S, Aggarwal G, et al. Novel class discovery without forgetting[C]//European Conference on Computer Vision. Cham: Springer Nature Switzerland, 2022: 570-586.

[2] Roy S, Liu M, Zhong Z, et al. Class-incremental novel class discovery[C]//European Conference on Computer Vision. Cham: Springer Nature Switzerland, 2022: 317-333.

[3] Liu M, Roy S, Zhong Z, et al. Large-scale pre-trained models are surprisingly strong in incremental novel class discovery[C]//International Conference on Pattern Recognition. Cham: Springer Nature Switzerland, 2024: 126-142.

**Experimental Designs Or Analyses:**

Yes.

**Methods And Evaluation Criteria:**

Yes.

**Other Comments Or Suggestions:**

NA

**Other Strengths And Weaknesses:**

Strengths:
1) the paper is well-written and well-presented. It is easy to read.
2) the proposed Bayesian C-GCD framework is novel, interesting, and highly generalizable.
3) the improvements on the evaluated benchmarks look promising and significant.
4) the abaltion study is sufficient.


Weakness:
1) the benchmarks used for evaluation are significantly limited. The proposed method is based on pre-trained DINO. Note that CIFAR, ImageNet, CUB, are all used in the representation learning of DINO. Although DINO does not use labels to supervise the model, **nearly none of  the "novel" classes the model meets at each incremental session are really novel** because these concepts, or even visual content are senn by the encoder during its pre-training. Therefore, I cannot hold high confidence about the effectiveness of the proposed method with current evaluation. Evalution on truely novel benchmarks are needed.
2) Insufficient discussion and comparison with related work. iNCD methods should be adapted and compared.
3) dependence on Initial Clustering Quality: The performance of VB-CGCD relies heavily on the quality of the initial clustering used to generate pseudo-labels. Poor clustering results, especially in high-dimensional spaces, could lead to inaccurate pseudo-labels, which may propagate errors and degrade the model's performance over time.

**Questions For Authors:**

Q1: Sensitivity to Hyperparameters: The method involves several hyperparameters, such as the regularization term and the early stopping threshold. The paper does not provide a detailed sensitivity analysis, leaving it unclear how robust the method is to variations in these hyperparameters across different datasets or tasks.

Q2: How would the method behave in other modalities (non-image data)?

**Relation To Broader Scientific Literature:**

C-GCD methods can be used to discover novel classes/concepts, which can benefits a spectrum of related fields, such as tissue discoevry and protein discovery.

**Theoretical Claims:**

Yes.

---

> ### Author Rebuttal · Authors · 2025-04-01
>
> ## Weakness:
>
> >1.
>
> We acknowledge that DINO-ViT-B16 was pre-trained on the ImageNet dataset, though its training was based on self-supervised learning using unlabeled data, and it was not trained on CIFAR or CUB. Since all our baselines including HAPPY (SOTA) utilize DINO for feature extraction, VB-CGCD also uses DINO to ensure a fair comparison with other methods. Additionally, to further assess the generalizability of our approach, we have conducted experiments on Stanford Cars, FGVC-Aircraft, CORE50, and Food-101 datasets.
>
> |Datasets|S0|S1|||S2|||S3|||S4|||S5|||
> |-|-|-|-|-|-|-|-|-|-|-|-|-|-|-|-|-|
> ||All|All|Old|New|All|Old|New|All|Old|New|All|Old|New|All|Old|New|
> |StanfordCars|72.55|69.41|71.06|45.38|67.08|67.75|56.46|65.45|66.18|53.51|62.35|63.77|37.68|60.92|61.22|55.39|
> |FGVCAircraft|82.32|76.23|79.22|55.47|74.95|75.28|72.34|70.15|71.91|54.51|65.13|67.84|43.67|60.66|62.64|58.03|
> |Core50|99.92|96.50|99.92|79.57|91.11|95.95|61.40|88.91|91.11|73.29|84.93|88.80|54.12|82.28|84.84|58.85|
> |Food101|84.78|81.67|81.80|81.04|79.51|79.87|77.32|78.25|77.85|81.04|75.41|76.30|68.2|73.30|73.76|69.12|
>
> Compared to other baselines, VB-CGCD outperforms them by a decent margin in terms of average overall accuracy, as shown below:
> |Methods|StanfordCar|FGVCAircraft|
> |-|-|-|
> |MetaGCD|54.67|47.16|
> |HAPPY|62.79|53.10|
> |VB-CGCD|65.04|69.42|
>
> > 2.
>
> Thank you for your valuable feedback. We will include a discussion on FRoST (Class-iNCD) in the related work section. FRoST employs a Gaussian distribution as the sampling distribution for its replay mechanism, incorporating samples from previously learned classes during training to mitigate forgetting. This represents a typical replay-based approach. In contrast, VB-CGCD does not utilize a replay mechanism but instead relies directly on prototypes for classification. The experimental comparison between FRoST and VB-CGCD is as follows:
>
> |Datasets|S0|S1|||S2|||S3|||S4|||S5|||
> |-|-|-|-|-|-|-|-|-|-|-|-|-|-|-|-|-|
> ||All|All|Old|New|All|Old|New|All|Old|New|All|Old|New|All|Old|New|
> |C100|FRoST|90.36|76.87|79.58|63.30|65.31|68.88|43.90|58.01|61.09|36.50|49.27|50.90|36.20|48.03|48.17|46.80|
> |TINY|FRoST|85.86|75.15|78.56|58.10|65.64|67.83|52.50|51.32|54.31|30.40|48.22|52.14|16.90|40.15|42.73|16.90|
> |IN100|FRoST|96.20|87.50|92.96|60.20|79.63|83.37|57.20|76.78|77.00|75.20|66.18|68.65|46.40|63.82|66.40|40.60|
> |CUB|FRoST|90.26|77.03|83.95|43.53|50.77|53.46|34.33|46.42|49.31|26.09|39.40|41.47|23.08|34.55|35.12|29.45|
>
>
> In terms of final overall accuracy, VB-CGCD outperforms FRoST by an average of 30.76%, demonstrating its superior performance.
>
> >3.
>
> In unsupervised scenarios, most methods are influenced by the performance of the clustering algorithm. Our strategy is to minimize errors introduced by the clustering algorithm from the classifier's perspective to enhance performance. To ensure fair comparisons with other methods, we have employed the fundamental k-means algorithm. Of course, our approach can also be integrated with more advanced clustering algorithms to further improve performance.
>
> ## Q&A
>
> >1.
>
> Across all datasets and tasks, we employed a consistent set of hyperparameters, including learning rate and training epochs. Specifically, the fine-tuning coefficient λ was uniformly set to 0.001.
> Regarding the early stopping strategy, we posit that equalizing the average covariance between new and old classes effectively mitigates class bias. Therefore, we monitor the covariance of both new and old classes during training and halt training when they are equal, i.e., when R=0. We have conducted a sensitivity analysis on the R. Please refer to Response 4 of Reviewer 2.
>
> >2.
>
> Since our method performs density estimation solely on features, it can be broadly applied to various modalities. We conducted experiments on audio and text classification tasks.
> For audio we evaluated our method on esc50, speechcommands, and audiomnist datasets. For text, we evaluated stackoverflow, and clinc.
>
> |Datasets|S0|S1|||S2|||S3|||S4|||S5|||
> |-|-|-|-|-|-|-|-|-|-|-|-|-|-|-|-|-|
> ||All|All|Old|New|All|Old|New|All|Old|New|All|Old|New|All|Old|New|
> |ESC50|72.98|75.10|72.03|96.66|78.05|74.68|100.0|78.68|78.05|82.92|80.33|78.05|100.0|78.0|78.93|70.45|
> |Speechcommands|85.25|80.43|82.37|72.11|78.27|79.13|72.67|71.75|76.92|45.38|70.59|70.54|71.01|69.18|69.52|66.44|
> |AUDIO-MNIST|96.22|96.40|96.22|97.26|96.94|96.40|100.0|97.05|96.74|98.85|96.94|96.71|98.68|96.4|96.78|93.75|
> |Clinc|98.66|94.22|98.22|74.22|92.25|94.0|81.77|90.83|91.87|83.55|90.71|90.66|91.11|88.66|90.32|73.77|
> |Stackoverflow|89.0|86.83|87.2|85.0|86.28|86.66|84.0|84.75|85.57|79.0|84.88|84.25|90.0|84.8|84.77|85.0|
>
> The experimental results indicate that VB-CGCD demonstrates good performance in these tasks as well, further validating its generalizability and robustness to other modalities.

---

### Official Review · Reviewer_CRqZ · 2025-03-11

**Overall Recommendation:** 4

**Summary:**

This paper studies the task of Continual Generalized Category Discovery (CGCD). It analyzes C-GCD’s forgetting dynamics through a Bayesian lens, revealing that covariance misalignment between old and new classes drives performance degradation. To solve these issues, this paper proposes Variational Bayes C-GCD (VB-CGCD) integrates variational inference with covariance-aware nearest-class-mean classification. VB-CGCD adaptively aligns class distributions while suppressing pseudo-label noise via stochastic variational updates.

**Claims And Evidence:**

Yes.

**Essential References Not Discussed:**

Please discuss the recent work of C-GCD Happy [R1] in the Related Work.

References:
[R1]. Happy: A Debiased Learning Framework for Continual Generalized Category Discovery. NeurIPS 2024.

**Experimental Designs Or Analyses:**

Yes.

**Methods And Evaluation Criteria:**

Yes.

**Other Comments Or Suggestions:**

No.

**Other Strengths And Weaknesses:**

Strengths:
1. This paper is well-motivated and easy to follow.
2. The writing and the diagram are clear. I appreciate Figure 1.
3. The proposed variational Bayes-based method is novel to the community of category discovery.
4. The performance gains are remarkable.

Weakness:
1. Discussion of related work Happy [R1] is missing in the Related Work.
2. Why are some results of Happy in Table1 different from the reported results in the original paper [R1]?
3. Please include a brief introduction of the method (i.e., spirit, pipeline, etc) in the caption of figure 1.
4. The experiments of hyperparameters should be added, including $\epsilon$ in early stopping.
5. Could the authors provide some experiments that support the claim that self-supervised loss during continual learning introduces feature drift?

References:
[R1]. Happy: A Debiased Learning Framework for Continual Generalized Category Discovery. NeurIPS 2024.

**Questions For Authors:**

See Other Strengths And Weaknesses.

**Relation To Broader Scientific Literature:**

This paper makes novel contributions of bayes variational inference for continual generalized discovery.

**Theoretical Claims:**

This paper derives the method from the perspective of variational bayes, which is clear.

---

> ### Author Rebuttal · Authors · 2025-04-01
>
> >1.
>
> We appreciate the reviewer’s valuable feedback. We have regarded HAPPY as an important SOTA but inadvertently missed its discussion in the related work section. We will include the discussion of HAPPY in the revised version, with a particular emphasis on its technique of using Gaussian distributions for prototype sampling, to provide a more comprehensive overview of the relevant work.
>
> >2.
>
> We reproduced the experimental results of HAPPY and found that our reproduced results outperform those reported in the original paper. We reported the reproduced results in our paper.
>
> >3.
>
> We will update the overall process in the caption of Table 1 as follows:
> "(a) Offline Stage: Labeled data is used to fine-tune the backbone network. After fine-tuning, features are extracted, and variational inference is performed to obtain and store the class distribution.
> (b) Online Stage: Unlabeled data is first processed by the feature network to extract features, which are then clustered to assign pseudo-labels. These pseudo-labels are combined with a first round of variational inference to obtain an approximate distribution, then combined with the stored class distribution from the previous session for re-labeling."
>
> >4.
>
> Across all datasets, we employed the same hyperparameter configuration, including conventional parameters such as epochs and learning rate, as detailed in Appendix C.3.
> Regarding the fine-tuning hyperparameter λ, we found that the contribution of self-supervision to performance is almost negligible. This is primarily because, compared to self-supervised learning on extremely large-scale pretraining datasets, the fine-tuning dataset has a limited impact on performance, with the cross-entropy loss playing a more crucial role. Therefore, we set λ to a very small value of 0.001.
>
> As for the early stopping hyperparameter, as discussed in Figure 2, we argue that ensuring equal average covariance between new and old classes can effectively mitigate class bias. Consequently, our early stopping strategy dictates training to be stopped once the covariances of new and old classes are equal (i.e., when R equals 0).
> We conducted experiments on the CIFAR 100 dataset, focusing on the parameter R, adjusting its value to achieve different trade-offs between model plasticity and stability.
>
> #### Comparisons in overall accuracy as R varies
>
> ||S1|S2|S3|S4|S5|
> |----|----|----|----|----|----|
> |R=3|85\.55|82\.2|79\.38|77\.74|75\.61|
> |R=2|86\.85|83\.3|80\.2|78\.41|76\.13|
> |R=1|88\.4|85\.9|83\.41|82\.57|81\.11|
> |R=0|88\.4|86\.0|83\.61|82\.72|81\.23|
> |R=\-1|87\.03|83\.75|81\.31|80\.18|77\.91|
> |R=\-2|87\.01|83\.71|81\.23|79\.94|77\.71|
>
> #### Comparisons in the accuracy of old classes as R varies
> ||S1|S2|S3|S4|S5|
> |----|----|----|----|----|----|
> |R=3|91\.62|85\.31|81\.98|79\.17|77\.44|
> |R=2|91\.32|86\.63|83\.15|80\.01|78\.11|
> |R=1|90\.18|87\.2|84\.64|82\.81|81\.92|
> |R=0|89\.46|87\.06|84\.75|82\.85|81\.84|
> |R=\-1|87\.34|84\.2|81\.91|79\.57|77\.91|
> |R=\-2|87\.30|84\.10|81\.82|79\.31|77\.56|
>
> #### Comparisons in the accuracy of new classes as R varies
> ||S1|S2|S3|S4|S5|
> |----|-|-|---|-|-|
> |R=3|55\.2|63\.5|61\.2|66\.3|59\.1|
> |R=2|64\.5|63\.3|59\.5|65\.6|58\.3|
> |R=1|79\.5|78\.1|74\.8|80\.7|73\.8|
> |R=0|83\.1|79\.6|75\.6|81\.7|75\.7|
> |R=\-1|85\.5|81\.1|77\.1|85\.1|77\.9|
> |R=\-2|85\.6|81\.4|77\.1|85\.0|79\.0|
>
> The experimental results indicate that the parameter R serves to balance the accuracy between new and old classes, with the overall accuracy being highest when R equals 0. We will include these details in the updated version of the paper.
>
> >5.
>
> In continual learning, new data is introduced at each session. If the backbone network is continuously fine-tuned, the features extracted from the same input using different versions of the network (e.g., h₁ and h₂) will inevitably differ, i.e., h₁(x) ≠ h₂(x). We designed an experiment on CIFAR-100, where we continuously perform self-supervised fine-tuning using data from each session. Subsequently, we evaluate feature drift by conducting supervised learning.
>
> |SSL\-finetuned|overall|S0|S1|S2|S3|S4|S5|
> |--|-|--|---|--|--|--|--|
> |S0|85\.37|85\.3|83\.0|84\.9|85\.2|90\.1|84\.0|
> |S1|84\.55|84\.95|81\.9|83\.4|84\.8|86\.7|84\.0|
> |S2|83\.84|83\.94|82\.0|82\.28|82\.4|88\.0|83\.6|
> |S3|85\.02|85\.34|81\.8|84\.1|84\.6|89\.1|84\.0|
> |S4|84\.45|84\.76|82\.5|83\.5|83\.5|88\.1|83\.2|
> |S5|85\.01|84\.99|81\.9|84\.6|85\.6|88\.6|84\.4|
>
> The experimental results indicate that as fine-tuning progresses, the accuracy of each session, as well as the overall accuracy, fluctuates within a 2% range, demonstrating the presence of feature drift. Moreover, this drift does not have a consistently positive impact on accuracy and does not lead to significant performance improvements. Since our classification relies on storing the distribution of old classes, any drift may result in inaccurate classifications. Therefore, we chose to avoid continuously fine-tuning the pre-trained network.

---

### Official Review · Reviewer_vXPB · 2025-03-12

**Overall Recommendation:** 2

**Summary:**

This manuscript investigates the problem of Continual Generalized Category Discovery, addressing the challenges of mixed new and old categories and high uncertainty in unlabeled data under a continual learning setting. The authors propose a new variational Bayesian framework that utilizes offline fine-tuning and self-correcting re-labeling to mitigate label bias and label noise. By employing covariance-aware early stopping, the framework balances new-category plasticity and old-category stability.

**Claims And Evidence:**

Yes

**Essential References Not Discussed:**

No

**Experimental Designs Or Analyses:**

Yes

**Methods And Evaluation Criteria:**

Yes

**Other Comments Or Suggestions:**

None

**Other Strengths And Weaknesses:**

**Strengths**:

1.The paper is well written, with strong readability and logical clarity.

2.The newly designed framework demonstrates impressive performance, achieving state-of-the-art results.



**Weaknesses:**

1.The experimental design is not sufficiently comprehensive. The ablation study only covers the removal of entire modules but does not analyze specific design components within those modules. For instance, the paper mentions using Mahalanobis similarity to account for covariance structures and mitigate the curse of dimensionality. However, the benefit of introducing Mahalanobis similarity into the proposed framework has not been clearly demonstrated.

2.Although the paper states that existing methods can effectively estimate the number of new categories when it is unknown, there are no experiments or analyses for scenarios in which the number of new categories in each session is indeed unknown. Such a setting would be crucial to validate the effectiveness of the proposed framework in real-world conditions.

3.The framework reduces interference by distinguishing between new and old categories, offering an improvement over purely distance-based methods. However, the paper lacks a more in-depth comparison with open-set recognition and GCD approaches that also differentiate known from unknown classes. (e.g. Extreme Value Theory in open-set recognition and DPN[1] in GCD )

I would be happy to increase my score if the authors can resolve all concerns.

[1] Generalized Category Discovery with Decoupled Prototypical Network

**Questions For Authors:**

Please see weaknesses.

**Relation To Broader Scientific Literature:**

No

**Theoretical Claims:**

Yes

---

> ### Author Rebuttal · Authors · 2025-04-01
>
> >1.
>
> We conducted an ablation study on the benefit of introducing Mahalanobis distance, and the results obtained by replacing the Mahalanobis with the Euclidean distance are as follows:
> Datasets|Methods|All\(S0\)|All\(S5\)|Mf|Md
> --|---------|---|---|----|---
> C100|w/oMahalanobis|90\.52|76\.06|10\.1|74\.96
> Tiny|w/oMahalanobis|87\.64|70\.33|9\.7|67\.5
> IN100|w/oMahalanobis|93\.6|85\.06|8\.04|86\.56
> CUB|w/oMahalanobis|83\.78|55\.79|0\.82|28\.15
>
> Compared to VB-CGCD with Mahalanobis distance, the overall final accuracy decreased by approximately 5.56 on average. This is because Euclidean distance, being a special case of Mahalanobis, is susceptible to collapse in high-dimensional spaces. By incorporating covariance information, Mahalanobis distance effectively mitigates this issue, thereby achieving a more optimal classification boundary. We appreciate your valuable feedback and will incorporate this aspect into the revised version of our paper.
>
> > 2.
>
> Since our proposed approach primarily serves as an incremental learning classifier and is decoupled from the clustering module, it allows integration with any clustering method (including those that estimate the number of categories).
> As mentioned in the paper, there are many off-the-shelf methods [1][2] available for estimating the number of categories.
> Nonetheless, we employed the classic silhouette score to implement a method for estimating the number of unknown classes, which we integrated into VB-CGCD.
> The results are as follows:
> |Datasets|S0|S1|||S2|||S3|||S4|||S5|||
> |--|----|----|----|----|----|----|----|----|----|----|----|----|----|----|----|----|
> ||All|All|Old|New|All|Old|New|All|Old|New|All|Old|New|All|Old|New|
> |C100|91\.68|82\.48|87\.2|58\.9|79\.08|82\.31|59\.7|76\.25|78\.90|57\.7|75\.24|76\.07|68\.6|73\.13|74\.94|56\.8|
> |Tiny|88\.32|85\.55|86\.82|79\.2|82\.41|84\.01|72\.8|79\.87|81\.41|69\.1|77\.20|78\.82|64\.2|74\.77|76\.34|60\.6|
> |IN100|94\.32|91\.93|93\.12|86\.0|89\.45|90\.43|83\.6|88\.6|88\.28|90\.8|86\.11|87\.62|74\.0|84\.56|85\.08|79\.8|
> |CUB|85\.72|79\.98|83\.68|60\.91|65\.98|67\.42|57\.41|63\.91|65\.94|49\.47|57\.54|58\.24|51\.76|54\.40|57\.54|25\.61|
>
> Due to errors in estimating the number of unknown classes, the clustering error is exacerbated, ultimately leading to an average overall accuracy reduction of approximately 5.66. Nonethless, it demonstrates the scalability of our method to handle scenarios with an unknown number of types, and our performance still outperforms the SOTA method, HAPPY, which also uses the silhouette score.
>
> |Methods|All|Old|New|
> |- |--- |--- |--- |
> |HAPPY|68.80|72.40|45.74|
> |VB-CGCD|77.23|79.88|60.34|
>
>  [1] DeepDPM: Deep Clustering with an Unknown Number of Clusters. Meitar Ronen, Shahaf E. Finder, Oren Freifeld. CVPR 2022.
> [2] PromptCCD: Learning Gaussian Mixture Prompt Pool for Continual Category Discovery. Fernando Julio Cendra, Bingchen Zhao, Kai Han. ECCV 2024.
>
> > 3.
>
> Our approach distinguishes between known and unknown classes by approximately estimating the unknown distribution, thereby effectively reducing the interference of known classes on the learning of new ones—a strategy that is similar to certain open-set methods. However, there are two key differences between GCD and CGCD: 1) GCD does not need to address the forgetting problem inherent in continual learning; 2) In the unsupervised learning phase of CGCD, supervised data is inaccessible, making it unsuitable for semi-supervised approaches. For example, during training, DPN relies on using L_{known} as a part of the loss function, which renders it inapplicable in CGCD scenarios. Moreover, methodologically, DPN uses the mean as its prototype; however, in high-dimensional spaces, this approach is prone to feature collapse. In contrast, utilizing a Gaussian distribution is more robust. We will further emphasize this distinction in the related work section.
>
> Additionally, we evaluated VB-CGCD on the same datasets which were evaluated in DPN ‘s paper, and the results are shown below.
>
> |        |   banking | |   |   stackoverflow  | |  |   clinc     | | |
> |-------|--------|-------------|---------------------|----------------------|---------------------|---------------------------|---------------------|---------------------|--------------------|
> |        |   All       |   Known           |   Novel     |   All       |   Known     |   Novel     |   All       |   Known     |   Novel    |
> |   DPN        |   72\.96    |   80\.93          |   48\.60    |   84\.23    |   85\.29    |   81\.07    |   89\.06    |   92\.97    |   77\.54   |
> |   VB\-CGCD   |   75\.55    |   82\.88          |   53\.15    |   84\.2     |   83\.73    |   85\.6     |   90\.97    |   95\.16    |   78\.19   |
>
> The experimental results indicate that VB-CGCD achieves performance comparable to DPN. However, it is important to note that VB-CGCD does not need access to any labeled data during the unsupervised learning phase, which is a distinct difference from DPN.

---

### Decision · Program_Chairs · 2025-05-01

**Decision:**

Accept (poster)

**Comment:**

This paper introduces a variational Bayes-based framework for Continual Generalized Category Discovery (C-GCD), designed to address the challenges of incrementally learning new classes from unlabeled data streams while retaining knowledge of old, previously-seen classes. The authors analyze the forgetting dynamics in C-GCD from a Bayesian perspective and identify covariance misalignment between old and new classes as a key driver of performance degradation. Based on this insight, they introduce VB-CGCD, which integrates variational inference of Gaussian parameters and covariance-aware nearest-class-mean classification. Experimental results on standard benchmarks demonstrate that VB-CGCD significantly outperforms state-of-the-art methods. Of particular note is the performance of the proposed approach on a benchmark with only 10% labeled data in the first, offline supervised training session.

Reviewer opinion of the paper was initially mixed. Reviewers appreciated the readability and clarity of the technical narrative, the impressive performance improvement over baselines on standard benchmarks, and the novelty of the Bayesian C-GCD framework. On the other hand, they raised concerns over insufficiently comprehensive experiments, limited discussion of related work on incremental Novel Class Discovery, and the dependence on known number of classes on the quality of clustering.

The authors engaged with all reviewer concerns in their rebuttal, providing additional results and promising to incorporate more comprehensive discussion of related work and new results in the revised manuscript. Given that the standard practice in GCD and CGCD is to assume a known number of classes (and relegate estimation of this to a generic clustering algorithm), this criticism of the work is not well-founded -- although the authors are encouraged to include the comparison with HAPPY using silhouette score in the revised manuscript.

In terms of issues regarding coverage of the literature in the Related Work section, given page limits and need to narrowly zoom in on the real contributions for conference papers, it is very hard to range far afield and include everything even in the last year that has anything to do with variational inference, continual learning, or GCD. The authors articulated in rebuttal their motivations for excluding some of the mentioned works, and will update the Related Work to include some of the more relevant ones.

After addressing these critical issues in rebuttal, the consensus that remains is that this is a solid and novel contribution which yields excellent improvements over the state-of-the-art.